# New glacier thickness and bed topography maps for Svalbard

Ward van Pelt[1,*] and Thomas Frank[1,*]

[1]Department of Earth Sciences, Uppsala University, Sweden
[*]These authors contributed equally to this work.

**Correspondence:** Ward van Pelt (ward.van.pelt@geo.uu.se)

**Abstract.** Knowledge of the thickness, volume, and subglacial topography of glaciers is crucial for a range of glaciological, hydrological, and societal issues, including, e.g., studies on climate-warming induced glacier retreat and associated sea-level rise. This is not the least true for Svalbard, one of the fastest-warming places in the world. Here, we present new maps of the ice thickness and subglacial topography for every glacier on Svalbard. Using remotely sensed observations of surface height, ice velocity, rate of surface elevation change, and glacier boundaries in combination with a modeled mass balance product, we apply an inverse method that leverages state-of-the-art ice flow models to obtain the shape of the glacier bed. Specifically, we model large glaciers with the Parallel Ice Sheet Model (PISM) at 500 m resolution, while we resolve smaller mountain glaciers at 100 m resolution using the physics-informed deep learning-based Instructed Glacier Model (IGM). Actively surging glaciers are modeled using a perfect-plasticity model. We find a total glacier volume (excluding the island Kvitøya) of 6,800±238 km$^3$, corresponding to 16.3±0.6 mm sea level equivalent. Validation against thickness observations shows high statistical agreement, and the combination of the three methods is found to reduce uncertainties. We discuss the remaining sources of errors, differences from previous ice-thickness maps of the region, and future applications of our results.

## 1 Introduction

Glaciers outside the Greenland and Antarctic ice sheets currently account for about half of the total land ice contribution to sea level rise (Hugonnet et al., 2021). About 7% of the total glacier contribution to sea level rise between 1961/62 and 2015/16 came from glaciers in Svalbard and Jan Mayen, with an estimated 687 Gt of glacier mass loss (IPCC, 2023). Svalbard is experiencing among the fastest warming on the planet, as it experiences the direct impacts of amplified warming (Arctic Amplification) following the ongoing retreat of sea ice and associated radiation feedbacks (e.g. Serreze and Barry, 2011; Bintanja and Van der Linden, 2013; Cao et al., 2017). In response to a strong warming trend and a weak increase in precipitation, Svalbard glaciers have lost mass at a rate of 7±4 Gt a$^{-1}$ during 2000-2019 due to surface-atmosphere interactions, as expressed by the climatic mass balance (CMB), in addition to frontal ablation losses of 2±7 Gt a$^{-1}$ (Schuler et al., 2020). CMB predictions indicate an acceleration of mass loss with average CMB values below -50 Gt a$^{-1}$ in 2060 for the future emission scenarios RCP4.5 and RCP8.5 (Van Pelt et al., 2021). Based on historical data, structure-from-motion photogrammetry, and a space-for-time substitution, Geyman et al. (2022) estimated a doubling of glacier mass loss from 1936-2010 to 2010-2100 with an average thinning of -0.67 to -0.92 m yr$^{-1}$ in the latter period.

Knowledge of ice thickness and subglacial topography is relevant for many applications. The mean ice thickness and glacier volume provide estimates of fresh water storage on land. Glacier volume trends directly affect sea level rise (SLR), but also have an impact on future fresh water availability and management. Knowing the ice-free topography after glacier retreat gives insight in future landscape and coastlines, which is relevant for future marine, terrestrial, hydrological, ecosystem, and climate modeling studies. A necessity for simulating long-term glacier evolution is detailed knowledge of basal topography under the ice. Whereas a wealth of observational data of surface processes is available, the inaccessibility of the glacier bed complicates direct observations of subglacial topography. Measurement of distributed basal topography fields using ground-penetrating radar (GPR) is a laborious and expensive task. As a result, thickness observations exist for only 1-2 % of all glaciers worldwide (Gärtner-Roer et al., 2014; GlaThiDa Consortium, 2020).

Ice flow models simulate ice motion and changing ice geometry and are the common tool to study glacier mass and volume change in past, present and future climates (e.g. Goelzer et al., 2017; Rounce et al., 2023). A major source of uncertainty in glacier modelling, contributing to errors in sea level rise predictions, stems from difficulties in setting initial conditions in the present day that are needed as a starting point for forecasting runs. Knowledge of bed topography and friction is essential for accurate simulation of ice motion and thickening/thinning, but direct observations are scarce (Morlighem, 2022). This has stimulated the development of inverse methods to indirectly estimate the ice thickness distribution from much more abundant surface data, including surface height, mass balance and/or velocity. A range of inverse methods to produce ice thickness maps have been compared in Farinotti et al. (2017, 2021). Participating approaches ranged from point-based methods (e.g. Linsbauer et al., 2009) to fully-distributed methods (e.g. Van Pelt et al., 2013), and differed regarding the required input datasets (such as mass balance, velocity and surface height change), as well as the ice flow physics used.

The inverse methods used in this study are based on the iterative approach in Frank et al. (2023), which is inspired by the method in Van Pelt et al. (2013) and performs short forward simulations with an ice flow model around the time of collection of observational datasets of distributed velocity, surface height and its change, and mass balance. After every forward simulation (iteration) bed heights are adjusted to reduce mismatches of surface height change. On fast-flowing tide-water glaciers, basal friction is additionally optimized to reduce mismatches with surface velocity data. Using surface height and velocity errors to correct basal conditions has proven to be a fast method to converge to bed height and friction fields that, for the assumed ice flow physics, generate a glacier dynamic state that is consistent with observations (Frank et al., 2023). Uncertainties in observational datasets and model physics introduce errors in the bed, and to prevent "over-fitting" regularization is required, e.g. by smoothing input datasets. The inverse method itself does not introduce errors; in the hypothetical case of a perfect ice flow model and noise-free input datasets, the reconstructed basal conditions would approach reality. There is however a physical limit to the spatial detail that can be resolved as small-scale bed features do not yield any surface expression (Gudmundsson and Raymond, 2008). Advantages of the method in Frank et al. (2023) are that it can be used with any ice flow model and that the final state at the end of the inversion is a useful initial (spin-up) state for prognostic simulations, as the geometry and dynamics are consistent with surface observations.

In this study, different ice flow models are used to invert for bed topography of small land-terminating glaciers and to invert for bed topography and basal friction on large land-terminating and fast-flowing marine-terminating glaciers. For modelling

large land-terminating glaciers and tide-water glaciers on a 500-m resolution grid, a similar method as in Frank et al. (2023) is used, which employs the ice flow model Parallel Ice Sheet Model (PISM; www.pism.io; Bueler and Brown, 2009) that combines the shallow ice approximation (SIA) and shallow shelf approximation (SSA) to simulate internal deformation and sliding motion respectively. For modelling small land-terminating glaciers, we adopt the same approach as in a recent study by Frank and Van Pelt (2024), where the inverse method was applied to all glaciers in Norway and Sweden, using the machine-learning based Instructed Glacier Model (IGM; Jouvet and Cordonnier, 2023; Cook et al., 2023) as an ice flow model. The advantages of IGM over using traditional (shallow) ice flow models are 1) the ability to use a higher-order physics, which is particularly relevant for mountain glaciers, and 2) severely reduced numerical cost which enables simulations with high spatial resolution. In this study, IGM is used to model small land-terminating glaciers in Svalbard at a 100-m spatial resolution.

Svalbard is home to 1,567 glaciers (1,544 glaciers excluding Kvitøya) with a total area of 33,775 km$^2$ in ~2010 (Nuth et al., 2013). Of these glaciers, 186 (12%) are classified as tidewater glaciers, covering an area of 23,986 km$^2$, equivalent to 71% of the total glacier area (RGI Consortium, 2017, Randolph Glacier Inventory (RGI) version 6;). 103 glaciers in Svalbard have been reported to surge, and another 103 and 37 are respectively possibly or probably of surge-type (Sevestre and Benn, 2015). Several studies have previously quantified Svalbard's glacier volume and thickness using a wide range of methods. Volume-area scaling methods, often applied in global studies, have given volume estimates ranging from 4,000 km$^3$ (Ohmura, 2004) to 10,260 km$^3$ (Radić and Hock, 2010), and various estimates between these extremes (e.g. Hagen, 1993; Grinsted, 2013; Radić and Hock, 2013; Martín-Español et al., 2015). More recently, inverse methods have been used to reconstruct distributed ice thickness in global assessments (Farinotti et al., 2019; Millan et al., 2022) as well as in a dedicated regional study on Svalbard (Fürst et al., 2018b). Whereas Farinotti et al. (2019) presented a weighted average thickness distribution based on a set of thickness products produced using different methods, Millan et al. (2022) instead estimated thickness distribution using global high-resolution velocity data and assuming SIA-based ice flow physics and a Weertman sliding law. Millan et al. (2022) estimated Svalbard's glacier volume at 6,855 km$^3$ (excluding Kvitøya). Fürst et al. (2018b) used a two-step mass conservation method (Fürst et al., 2017) that locally calibrates ice viscosity using thickness observations in the Glacier Thickness Database (GlaThiDa; GlaThiDa Consortium, 2020). The method by Fürst et al. (2018b) hence locally assimilates the thickness data, and errors were shown to increase with distance to observation locations. Fürst et al. (2018b) found a volume estimate of 6,199 km$^3$, and a likely range of 5,200-7,300 km$^3$. The thickness results for Svalbard in Farinotti et al. (2019) are a copy of the results in Fürst et al. (2018b), which is version 1.0 of the Svalbard ice-free topography (SVIFT; Fürst et al., 2018a). A newer version 1.1 of SVIFT is also available at Fürst et al. (2018a), which shows a $\sim$ 20 % higher volume (7,370 km$^3$) than version 1.0.

We present a new thickness and bed height dataset for all glaciers in Svalbard, using a combination of inverse model results using IGM on small land-terminating glaciers (at 100-m resolution) and PISM on large land-terminating and tidewater glaciers (at 500-m resolution). Surging glaciers were modeled separately with a perfect-plasticity method instead, as time-stamp mismatches of the input datasets (e.g. DEM from 2010 and velocity map from 2017-2018) did not allow for accurate inversion using the Frank et al. (2023) method for glaciers with strong short-term changes in geometry and flow dynamics. In the following sections, we describe the input data (Section 2), introduce the inverse method (Section 3), present the bed

**Table 1.** Overview of the datasets used in the inverse method.

| Variable | Method/database | Orig. resolution | Time-frame | Source |
|---|---|---|---|---|
| Digital elevation model | Aerial photos | 20 m | 2009-2012 | NPI (2014) |
| Surface height change | ASTER & ArcticDEM | 100 m | 2010-2019 | Hugonnet et al. (2021) |
| Ice velocity | Landsat 8, Sentinel-2 & Sentinel-1 | 50 m | 2017-2018 | Millan et al. (2022) |
| Climatic mass balance | Energy balance - firn model (EBFM) | 1,000 m | 2010-2019 | Van Pelt et al. (2019) |
| Ice thickness | Glacier Thickness Database (GlaThiDa) v 3.1.0 | 966,408 data points | 1983-2016 | GlaThiDa Consortium (2020) |
| Frontal ablation | GlaThiDa & ITS_LIVE | Estimate per glacier | 2010-2020 | Kochtitzky et al. (2022) |
| Glacier outlines | Randolph Glacier Inventory 6.0 | - | 2000-2010 | RGI Consortium (2017) |

and thickness maps and compare them against existing products (Section 4), discuss uncertainties (Section 5), and present conclusions (Section 6).

## 2  Data

Various remote sensing and model-based datasets of surface conditions are used as "input" in the inverse method, including distributed maps of surface elevation, climatic mass balance, surface height change, glacier outlines, surface velocity, and
100 glacier-average frontal ablation. In addition, ice thickness observations are used for calibration and validation. The data is summarized in Table 1. Distributed maps of surface elevation, surface height change, velocity, climatic mass balance, thickness observations, and glacier outlines are shown in Fig. 1. For more details about the input datasets, the reader is referred to the data sources in Table 1. The main criteria for the selection of input datasets were: 1) performance in previous comparisons (when available), 2) the time-stamp, since data from a similar period were preferred, and 3) smoothness / spatial noise and missing
data. To support the selection of velocity and surface height change datasets we additionally performed tests forcing the inverse method with different products, i.e. Millan et al. (2022), Friedl et al. (2021) and NASA MEaSUREs ITS_LIVE (Gardner et al., 2023) for velocity, and Morris et al. (2020) and Hugonnet et al. (2021) for surface height change. This revealed the best performance against thickness data when using Millan et al. (2022) and Hugonnet et al. (2021) respectively (not shown). For surface heights, we chose to use the S0 Terrengmodel by the Norwegian Polar Institute (NPI, 2014), which is a 20-m resolution
digital elevation model (DEM), based on aerial photos between 2009-2012 and derived from subset models (5-m resolution) for regions in Svalbard. For glacier outlines, we used version 6.0 of the RGI outlines (instead of the newer version 7.0) based on the compatibility of the outline dataset with frontal ablation estimates in Kochtitzky et al. (2022). Differences between the RGI versions 6.0 and 7.0 are in the delineation of individual glaciers, the combined area and the total outline are the same in both versions (see http://www.glims.org/rgi_user_guide/regions/rgi07.html).

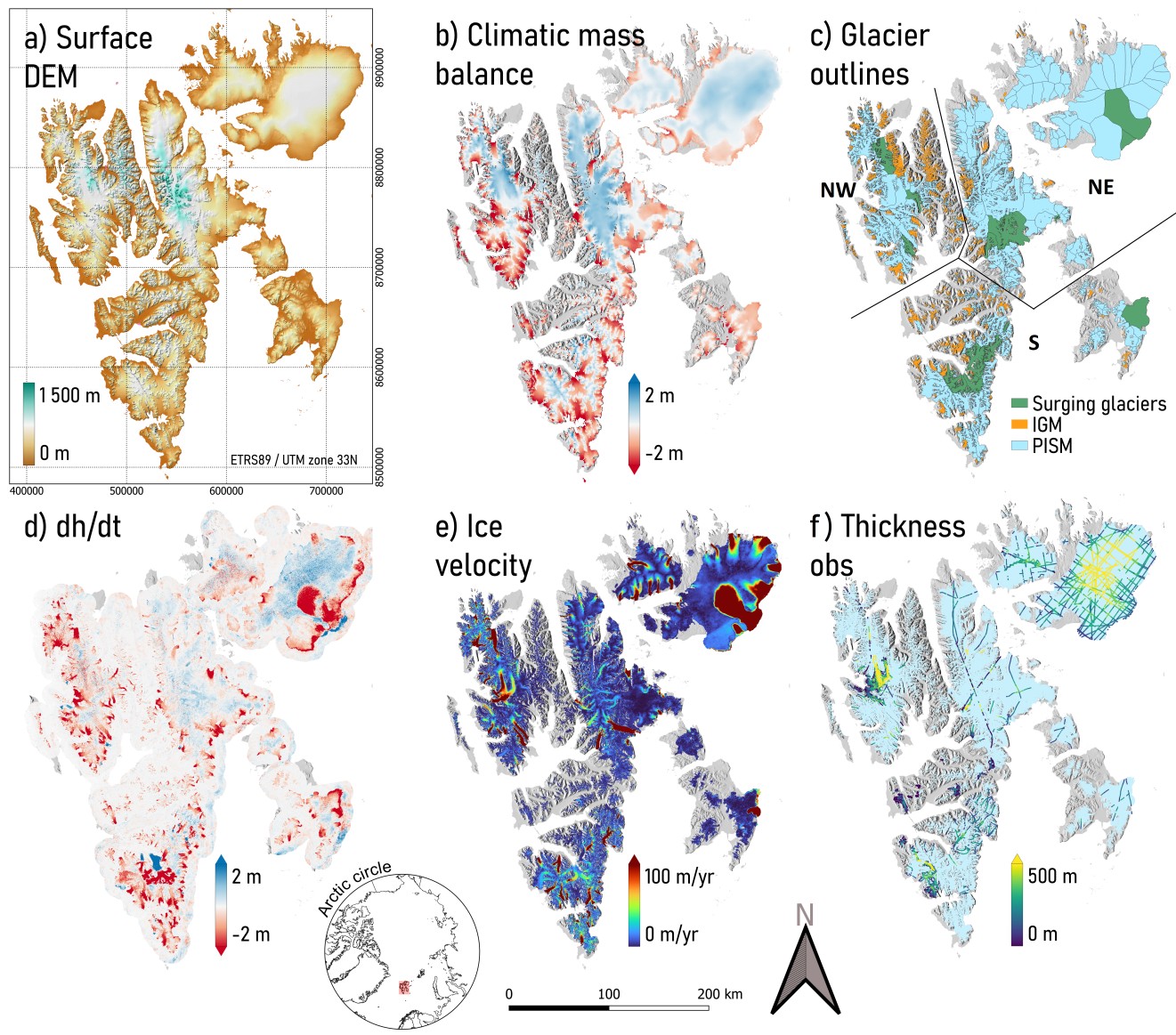

**Figure 1.** Overview maps of the input data sets used in the inverse modeling. Data sources and information are given in Table 1. The regions northwest (NW), northeast (NE) and southern Svalbard (S) are marked in c.

## 3  Methods

Three different approaches are used to generate thickness and bed maps for all glaciers in Svalbard. We split Svalbard's glaciers into three classes (see also Fig. 1c): 1) all glaciers that are smaller than 100 km$^2$ and not of tide-water and surge-type (Sevestre and Benn, 2015), 2) all glaciers that are larger than 100 km$^2$ and those smaller than 100 km$^2$ that are of tide-water or surge-type,

but not surging during 2015-2018 (Koch et al., 2023), 3) all glaciers that were reported to surge during 2015-2018. Glaciers in class 1 are modeled using the Instructed Glacier Model (IGM; Jouvet and Cordonnier, 2023) as in Frank and Van Pelt (2024) (Sect. 3.2). Glaciers in class 2 are modeled using the Parallel Ice Sheet Model (PISM; Bueler and Brown, 2009) as in Frank et al. (2023) (Sect. 3.1). Finally, ice thickness for glaciers in class 3 is estimated using the perfect-plasticity assumption (Nye, 1952). The rationale behind the grouping is that glaciers in class 1 can be modeled with higher resolution, higher-order physics, and low computational cost using the machine-learning model IGM. Large tide-water glaciers and ice caps, combining slow internally deforming sections with fast-flowing areas, are effectively modeled with PISM (Bueler and Brown, 2009). A simpler perfect-plasticity approach is needed for the surging glaciers in class 3, as mismatches in time-frames of input datasets (most prominently DEM, surface velocity & surface height change) would induce major errors when applying iterative inverse methods. One nuance to the three classes above is that all (small) glaciers in class 1 that are part of / connected to larger ice caps are modeled with PISM. This is to avoid thickness jumps at the ice divides. Furthermore, to avoid thickness jumps within ice caps between PISM-modeled and surging glaciers, experiments with PISM also include the surging glaciers as static entities with thicknesses based on the perfect-plasticity assumption. The three methods are described in more detail below.

## 3.1 Inversion using PISM

In preparation for the inversion, input datasets of the digital elevation model (DEM), surface height change, surface mass balance, and velocity were averaged/interpolated from their original grid (20-1,000 m resolution; Tab. 1) to the 500-m grid used by the ice flow model. Similarly, glacier outlines from the RGI were down-sampled onto the 500-m model grid to generate a mask separating glacier and glacier-free terrain.

The ice flow model PISM is used to perform iterative short (0.001 years) forward simulations of ice flow and geometric change for all glaciers in class 2, i.e. large ($>100$ km$^2$) glaciers and small quiescent surge-type glaciers. As in Frank et al. (2023), PISM uses the combined shallow-ice, shallow-shelf approximation (Bueler and Brown, 2009) to model both ice flow by internal deformation and sliding, the latter being described by a linear sliding law with spatially varying sliding coefficient $C$. A flow enhancement factor for the SIA ($SIA_\mathrm{e}$) is used, set here to 3 as in previous applications of PISM in Antarctica (Martin et al., 2011), Greenland (Bochow et al., 2023) and Iceland (Robinson, 2018). Ice temperature, and with that ice softness (3.1689e-24 Pa$^{-3}$ s$^{-1}$), are assumed to be constant, i.e. thermodynamics are not modeled. After every 0.001-year model run, modeled and observed surface height change ($\frac{dh_\mathrm{mod}}{dt}$ and $\frac{dh_\mathrm{obs}}{dt}$) are compared to calculate a misfit that is used to locally adjust the bed height $b$ before the next model run:

$$b_\mathrm{new} = b_\mathrm{old} - \beta \left( \frac{dh_\mathrm{mod}}{dt} - \frac{dh_\mathrm{obs}}{dt} \right) \tag{1}$$

where $\beta$ is a coefficient, set here to 0.25. Following Frank et al. (2023) we apply a simultaneous correction of the surface height, yet of opposite sign and a magnitude that is $\theta$ times the bed height misfit. The surface adjustments were previously found to stabilize the inversion in places where the ice flow model is not well able to simulate observed flow patterns e.g. because of simplifying assumptions in the stress balance equations (Frank et al., 2023). To avoid major surface height anomalies relative to the DEM, e.g. when starting from a strongly biased initial bed, we apply a one-time correction to the surface height map

after 400 iterations. During this correction, a map of surface height deviations relative to the DEM is computed and smoothed with a Gaussian filter (using four standard deviations for the Gaussian kernel); the resulting map is subtracted from the surface height map. Similar to Frank et al. (2023) we update basal friction (by modifying the sliding coefficient $C$). The initial friction field is derived from the linear sliding law $C = \left| \frac{\tau_d u_{\text{thres}}}{u_{\text{obs}}} \right|$, where $\tau_d$ is the driving stress, $u_{\text{thres}}$ is a threshold velocity (1 m s$^{-1}$), and $u_{\text{obs}}$ is the observed ice velocity (Bueler and Brown, 2009). Based on test runs, we found the best performance (lowest thickness errors) when updating $C$ only once after 400 model iterations. The inverse experiment uses a total of 800 iterations (bed height corrections). The initial bed at the start of the first model iteration $b_{\text{init}}$ is set to a bed that is estimated using the perfect-plasticity assumption (Nye, 1952; Li et al., 2012):

$$b_{\text{init}} = h - \frac{\tau}{\rho g \sin \alpha}, \tag{2}$$

where $h$ is the surface height, $\tau$ is a yield constant, $\rho$ is the ice density (900 kg m$^{-3}$), $g$ is the gravitational acceleration (9.8 m s$^{-2}$) and $\alpha$ is the absolute surface slope. For surface slopes smaller than $\alpha_{\text{min}}$, $\alpha = \alpha_{\text{min}}$, which is needed to avoid excessively large thickness values for low-sloping areas. Parameter values for $\alpha_{\text{min}}$ and $\tau$ were estimated based on calibration against thickness observations on surging glaciers, as described further in Sect. 3.3 below.

As in Frank et al. (2023), climatic mass balance per glacier is re-projected using a regression-based linear function of climatic mass balance with elevation. Similarly, we re-project surface height change using linear fitting against elevation. The linear regressions were previously found to increase the accuracy of reconstructed ice thicknesses as erroneous local spatial variations in the surface height change and velocity datasets no longer affect the thickness reconstruction (Frank and Van Pelt, 2024). Differencing of the climatic mass balance and surface height change results in the apparent mass balance (Farinotti et al., 2009), which is forced to sum to zero for every land-terminating glacier by applying spatially constant bias corrections per glacier. For tide-water glaciers, instead the glacier-summed apparent mass balance minus frontal ablation (Tab. 1; Kochtitzky et al., 2022) is enforced to equal zero. The above corrections assure mass conservation for every glacier, although compensating errors may occur, e.g. in the case of erroneous frontal ablation estimates resulting in a bias of the apparent mass balance. Despite the above measures to conserve mass, modeled glaciers often tend to become too thin at their fronts due to mass 'escaping' through the lateral boundaries set by the RGI outlines (Frank and Van Pelt, 2024). To compensate for this mass loss, we apply a fixed correction for all glaciers equal to $M_{\text{corr}}$. The positive apparent mass balance for tidewater glaciers together with a positive $M_{\text{corr}}$ commonly assure a positive mass flux (i.e. calving / frontal ablation) at the calving front. Hence, calving fronts do not retreat. They do not advance either since all mass that flows out of the outlines defined by the RGI dataset is instantly removed.

Frank et al. (2023) applied post-processing of thicknesses when modeled velocities in zones dominated by slow internal deformation flow were higher than observed even for $C \to \infty$. A different approach is applied here based on the logic that in zones where flow is controlled by internal deformation, the yield stress is an irrelevant parameter. We therefore introduced an observed velocity threshold $u_{\text{thres}} = 25$ m yr$^{-1}$ to identify regions where slow flow prevailed and no friction updates were applied.

Frank and Van Pelt (2024) previously found that ice thickness estimates improved by applying surface updates and mass balance corrections. With this in mind, we calibrated $\theta$ and $M_{\text{corr}}$, by searching for a minimum mean absolute error between

modeled and observed (GlaThiDa) ice thicknesses for all observed locations in Svalbard. Optimum values of $\theta = 0.4$ and $M_{\text{corr}} = 0.4$ m w.e. yr$^{-1}$ were found, yielding a mean absolute error (MAE) of 58.1 m. More statistics on the comparison with observations are given in Sect. 4.2. These statistics are after post-processing of thicknesses using a moving-average smoothing filter with a window size of 3 cells. This was found to give a reduction of the mean absolute error (-2.2 m), and an increase of
Pearson correlation (+0.014), relative to non-post-processed thicknesses. Bed heights are calculated by subtracting thicknesses from the DEM.

     Sensitivity tests were performed with a perturbed initial bed (zero ice thickness), magnitude of surface updates ($\theta = 0.2$ and $\theta = 0.8$), and mass balance correction ($M_{\text{corr}} = 0.2$ m w.e. yr$^{-1}$ and $M_{\text{corr}} = 0.6$ m w.e. yr$^{-1}$). Results are visualized in Figure 2 and show differences relative to the reference run with a perfect-plasticity-based bed, $\theta = 0.4$ and $M_{\text{corr}} = 0.4$ m
w.e. yr$^{-1}$. Figure 2 shows that $M_{\text{corr}}$ perturbation mostly affects lower elevation areas, whereas $\theta$ adjustments mainly impact slow-flowing high-elevation areas; this supports the choice of these two parameters for calibration. Impacts of perturbing $M_{\text{corr}}$ are increases of the MAE relative to the thickness observations of 1.3 m ($M_{\text{corr}} = 0.2$ m w.e. yr$^{-1}$) and 0.3 m ($M_{\text{corr}} = 0.6$ m w.e. yr$^{-1}$); perturbing $\theta$ yielded increases of the MAE of 2.5 m ($\theta = 0.2$) and 0.2 m ($\theta = 0.8$). Furthermore, perturbing $M_{\text{corr}}$ and $\theta$ introduce biases of the mean thickness of -10.6 ($M_{\text{corr}} =0.2$ m w.e. yr$^{-1}$), 7.3 ($M_{\text{corr}} =0.6$ m w.e. yr$^{-1}$), 5.1 ($\theta = 0.2$),
and -10.8 m ($\theta = 0.8$). The extreme case to start with no ice results in a weaker performance (e.g. 12.0 m increase in MAE), highlighting the importance of starting with a reasonable first guess of the bed topography. It is noteworthy that all perturbation experiments give a final bed at the end of the inversion that has a lower MAE than the initial (unperturbed) perfect-plasticity bed, which has an MAE equal to 77.5 m (compared with 58.1 m for our best run).

## 3.2   Inversion using IGM

The inversion for glaciers from class 1 follows a largely congruent workflow with the one above in that the principle is based on Frank et al. (2023) where bed updates (eq.(1)) and surface updates are executed iteratively. The main differences are the ice flow model (IGM v2.0.4 instead of PISM) and a few parameter and processing choices. The method is closely aligned with Frank and Van Pelt (2024). Note, therefore, that while we use IGM as a forward model, we do not use IGM´s built-in inversion as described by Jouvet (2022) which, in contrast to our method, assimilates thickness observations and relies on cost function
minimization.

     The spatial resolution is 100 m which the DEM and glacier outlines are down-sampled to. The DEM is furthermore smoothed in the ablation area with a two-sigma Gaussian filter; this strategy was found to be superior to not smoothing or to smoothing over the entire glacier area. The climatic mass balance for each glacier is downscaled from the original 1000 m resolution to 100 m by fitting an elevation dependent piece-wise linear function with two segments and a breakpoint at the ELA to the mass
balance product by Van Pelt et al. (2019) of a given glacier and glaciers within a buffer of 10 km. Taking neighboring glaciers into consideration is done to avoid poorly-constrained fits for small glaciers as a result of the coarse resolution of the original product. The apparent mass balance is calculated as above based on this new climatic mass balance and dh/dt.

     IGM (Jouvet and Cordonnier, 2023) is a physics-informed deep learning model that emulates higher-order ice flow while being computationally efficient. The underlying architecture is a Convolutional Neural Network (CNN) which is retrained as

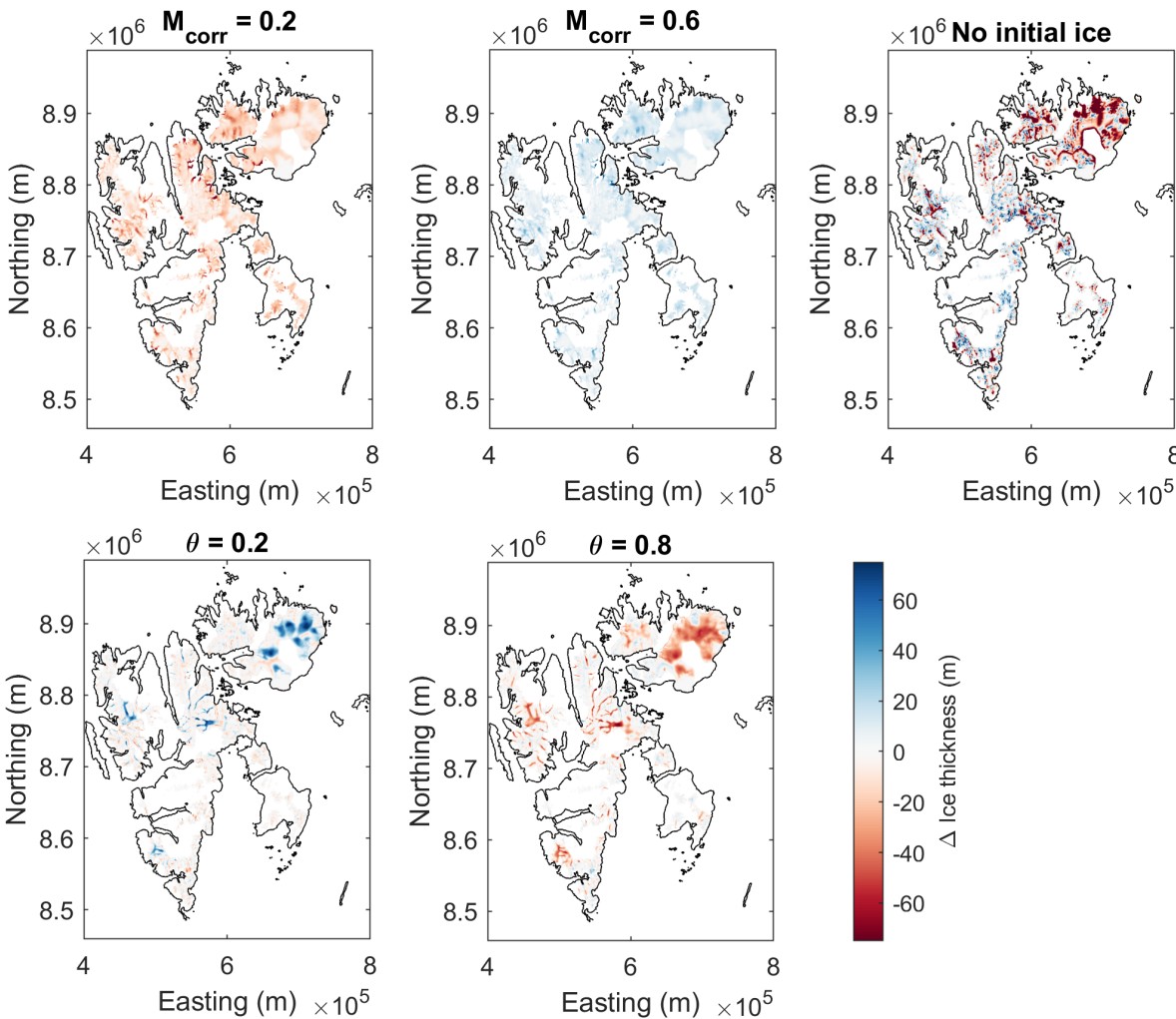

**Figure 2.** Sensitivity of PISM-modelled ice thickness for perturbed $M_{corr}$, $\theta$ and initial bed. Thickness differences are calculated by subtracting the thickness of the reference run ($M_{corr} = 0.2$ m w.e. yr$^{-1}$, $M_{corr} = 0.6$ m w.e. yr$^{-1}$, and perfect-plasticity-based bed) from the thickness of the perturbation experiment.

the model runs. This is achieved by comparing the solution of the CNN to that of an actual higher-order solver and updating the CNN weights based on that mismatch every 10th model iteration, ensuring a close alignment between the CNN and process model solutions. IGM includes a Weertman-type sliding law with a sliding coefficient c and it allows to set the ice viscosity parameter A. Calibration is done by finding one global value for A and c which minimizes the mean error to ice thickness observations. By not allowing A to exceed $A_{max} = 78$ MPa$^{-3}$a$^{-1}$ (the value corresponding to an ice temperature of 0°C) and

enforcing c=c$_{min}$=100 m MPa$^{-3}$a$^{-1}$ if A<A$_{max}$ (following a simplifying assumption that no basal sliding occurs for cold ice as

in Jouvet (2022) and Frank and Van Pelt (2024)), the calibration procedure yields the optimal parameters $A = 78$ MPa$^{-3}$a$^{-1}$, $c = 8000$ m MPa$^{-3}$a$^{-1}$. These values are applied uniformly to all glaciers in class 1.

The initial thickness field is obtained using a perfect plasticity approach (eq. (2)) with $\tau = 100$ kPA and $\alpha_{\mathrm{min}}= 0.04$. These perfect plasticity parameter values were selected based on sensitivity tests with IGM, and hence deviate from the ones used to generate the initial bed for glaciers in class 2 and the final bed for glaciers in class 3. Then, using IGM, 5000 model years are simulated during which bed (with $\beta=1$) and surface updates (with $\theta=0.25$) are applied. Whereas $\beta$ affects the magnitude of bed corrections and number of iterations needed, it hardly (if at all) influences the final bed; a too high value may however cause instabilities and values in PISM and IGM have been chosen accordingly. As in PISM, the value for $\theta$ in IGM has been optimized by minimizing discrepancies with thickness observations. In contrast to the PISM approach, basal friction is not updated but kept fixed. This follows from the assumption that there are smaller spatial variations in the basal friction fields of small mountain glaciers compared to large (tidewater) glaciers, meaning that one initial calibration of the spatially uniform c is sufficient. To account for mass escaping through the lateral glacier boundaries a different strategy than in the PISM approach is pursued, as in Frank and Van Pelt (2024). Specifically, after 3000 model years and for each glacier individually, the integrated apparent mass balance of those areas within the glacier mask that are ice-free (which is equal to the mass leakage rate) divided by the total glacier area is added to the specific apparent mass balance. Doing so, the mass leaking out on the lateral glacier boundaries is fed back to the glacier via a correction of the apparent mass balance. The final thickness field is obtained by interpolating gaps in the modeled thicknesses which may remain in the case of persistent mass leaking and applying a thickness-dependent Gaussian filter as in Frank and Van Pelt (2024).

### 3.3 Surging glaciers

Thickness estimation using iterative inverse methods as in Sect. 3.1 and 3.2 ideally uses input datasets of surface height, surface height change, velocity, mass balance and frontal ablation that represent the same point in time. In practice, accessible datasets will have different time stamps, introducing a source of error for inverse estimated thicknesses. Such errors are small for glaciers that are near steady-state or undergoing gradual change. Conversely, errors become considerable for glaciers that are undergoing rapid dynamic changes, e.g. in the event of surge initiation. In the latter case, a simpler method depending on fewer input datasets is desirable. Here, we apply the perfect-plasticity assumption to estimate thicknesses for 13 glaciers, including e.g. Basin-3, Negribreen and Tunabreen, that actively surged during 2015-2018, as identified by Koch et al. (2023). In the perfect plasticity assumption ice thickness is controlled primarily by the surface height (Eq.2). Since the DEM (2009-2012) was collected prior to the initiation of the surge for the selected glaciers, the thickness estimation is effectively based on the pre-surge glacier geometry. We regard this as an advantage as ice flow models in general are not well able to describe the strongly transient stress-state of actively surging glaciers. The application of the perfect-plasticity assumption is the same as when generating the initial bed in the PISM-based inversion (Sect. 3.1). To find optimum values of minimum slope $\alpha_{\mathrm{min}}$ and yield constant $\tau$ all combinations with $\alpha_{\mathrm{min}} = 0.010 : 0.001 : 0.040$ and $\tau = 0 : 2 : 100$ kPa were tested to find an optimum combination (lowest RMSE for all available thickness data on the 13 actively surging glaciers). This resulted in $\alpha_{\mathrm{min}} = 0.014$ and $\tau = 52$ kPa.

## 3.4 Combining the thickness datasets

The three inverse approaches (Sect. 3.1-3.3) generate distributed thickness and bed height datasets at different spatial resolutions: 100-m for the IGM-modelled glaciers and 500-m for both the PISM-modelled and the surging glaciers. To create a uniform combined map of ice thickness (and basal topography), results for the PISM-modelled and surging glaciers on the 500-m resolution grid have been re-projected to the finer 100-m resolution grid used by IGM using nearest-neighbor interpolation. Finally, to improve spatial detail of the outlines of the PISM-modelled and surging glaciers, glacier extent has been clipped to a 100-m resolution glacier mask extracted from the RGI dataset (RGI Consortium, 2017).

## 3.5 Estimating volume uncertainty

Given model complexity analytical error propagation of modelling errors is not feasible to estimate the uncertainty of the calculated ice volume for all glaciers. We instead adopt an alternative statistical method. The total volume $V$ of all glaciers is:

$$V = \bar{H} A, \tag{3}$$

where $\bar{H}$ is the mean ice thickness and $A$ is the area. Standard error propagation then implies that the standard error $\sigma_V$ results from errors in $\bar{H}$ and $A$ as follows:

$$\sigma_V = V \sqrt{\left(\frac{\sigma_A}{A}\right)^2 + \left(\frac{\sigma_{\bar{H}}}{\bar{H}}\right)^2}. \tag{4}$$

The term $\frac{\sigma_A}{A}$ is the relative area error resulting from uncertainty of outlines. Nuth et al. (2013) previously estimated this uncertainty 0.01-0.02 (1-2 %) for glaciers in Svalbard; we therefore assume an uncertainty of $\frac{\sigma_A}{A} = 0.015$ applies here. The term $\frac{\sigma_{\bar{H}}}{\bar{H}}$ is the relative mean thickness error. Through calibration of our inverse method, we effectively removed the bias between the average modeled and observed thickness, implying a negligible mean thickness error for the observed glaciers. This does not mean that average modeled thicknesses are bias-free at the Svalbard-wide scale, because of the smaller sample size of the observed glaciers relative to the total number of glaciers. In other words, a volume error may result from the fact that we calibrate against a finite sample of thickness data and use the same model setup also for glaciers without observations. To calculate $\sigma_{\bar{H}}$ we first calculate individual biases for all of the 169 glaciers in Svalbard with thickness observations in at least 10 model grid cells (on the 100x100 m grid), which gives values ranging from $-154$ to $163$ m, and a distribution of biases that is normally distributed according to a Lilliefors test. In the next step, we calculate the standard deviation of the 169 biases, giving 45.6 m, implying that if we calibrated against data from only one glacier, the mean modeled thickness would be off by between $-45.6$ and $+45.6$ m with a likelihood of 68 %. The range of biases narrows if we select more than one glacier for calibrating the model, and, following the same logic as is used to calculate a standard error of a mean, it can be found that dividing by the square root of the number of samples is required to calculate the remaining standard deviation for larger sets of glaciers used for calibration. Here, 169 glaciers were used for calibration, implying that the mean thickness error for all observed glaciers $\sigma_{\bar{H}}$ is found by dividing with the square root of the number of observed glaciers ($\sqrt{169}$), giving $\sigma_{\bar{H}} = 3.5$ m. With a mean observed thickness $\bar{H} = 257.2$ m, the relative thickness error $\frac{\sigma_{\bar{H}}}{\bar{H}}$ then becomes 0.014 (or 1.4%). As a result, we

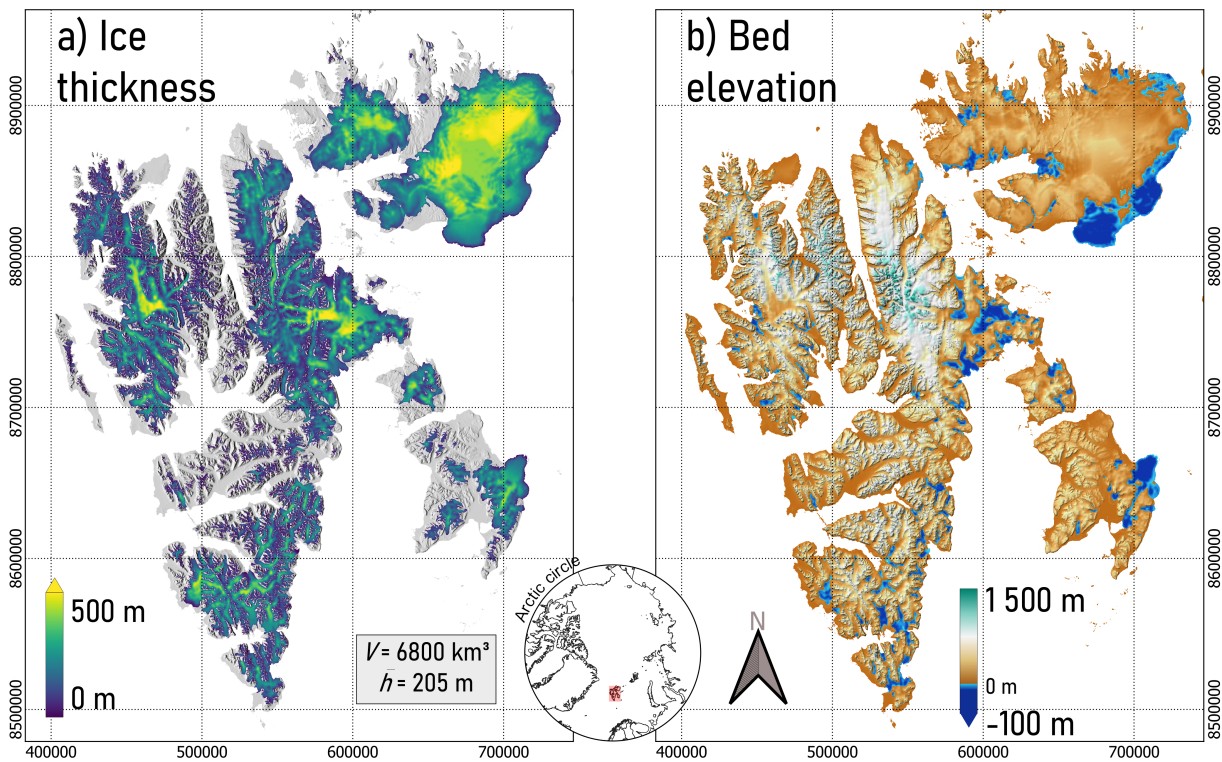

**Figure 3.** Ice thickness (a) and basal topography (b) for all glaciers in Svalbard (excluding Kvitøya).

find a (relative) standard error of volume $\frac{\sigma_V}{V}$ of 2.1 % from uncertainties in the area (outlines) and mean thickness; the 90-% confidence interval ($\pm 1.65 \frac{\sigma_V}{V}$) is hence $\pm 3.5$ %. Please note that the relative error of the volume and mean thickness is much smaller than the local (point) uncertainty of modeled thicknesses (the latter is quantified in Sect. 4.2).

# 4  Results & Discussion

## 295  4.1  Bed height, ice thickness and volume

Maps of ice thickness and bed topography, combining results from the three methods (Sect. 3), are shown in Figure 3. The mean thickness of all glaciers and ice caps in Svalbard, excluding Kvitøya, is estimated at 205 m. Ice volume equals 6,800 km$^3$, of which an estimated 315 km$^3$ (4.6 %) lies below sea level. Total volume uncertainty, with a 90% confidence interval, is estimated at $\pm 238$ km$^3$ ($\pm 3.5$ %; Sect. 3.5). Assuming an ice density of 917 kg m$^{-3}$, a seawater density of 1027 kg m$^{-3}$ and a global ocean area of $3.618 \times 10^8$ km$^2$ implies that the Svalbardian glaciers would raise global mean sea level by $16.3 \pm 0.6$ mm if they melted completely. The largest ice thicknesses are found on Austfonna (Nordaustlandet), Holtedahlfonna (northwest

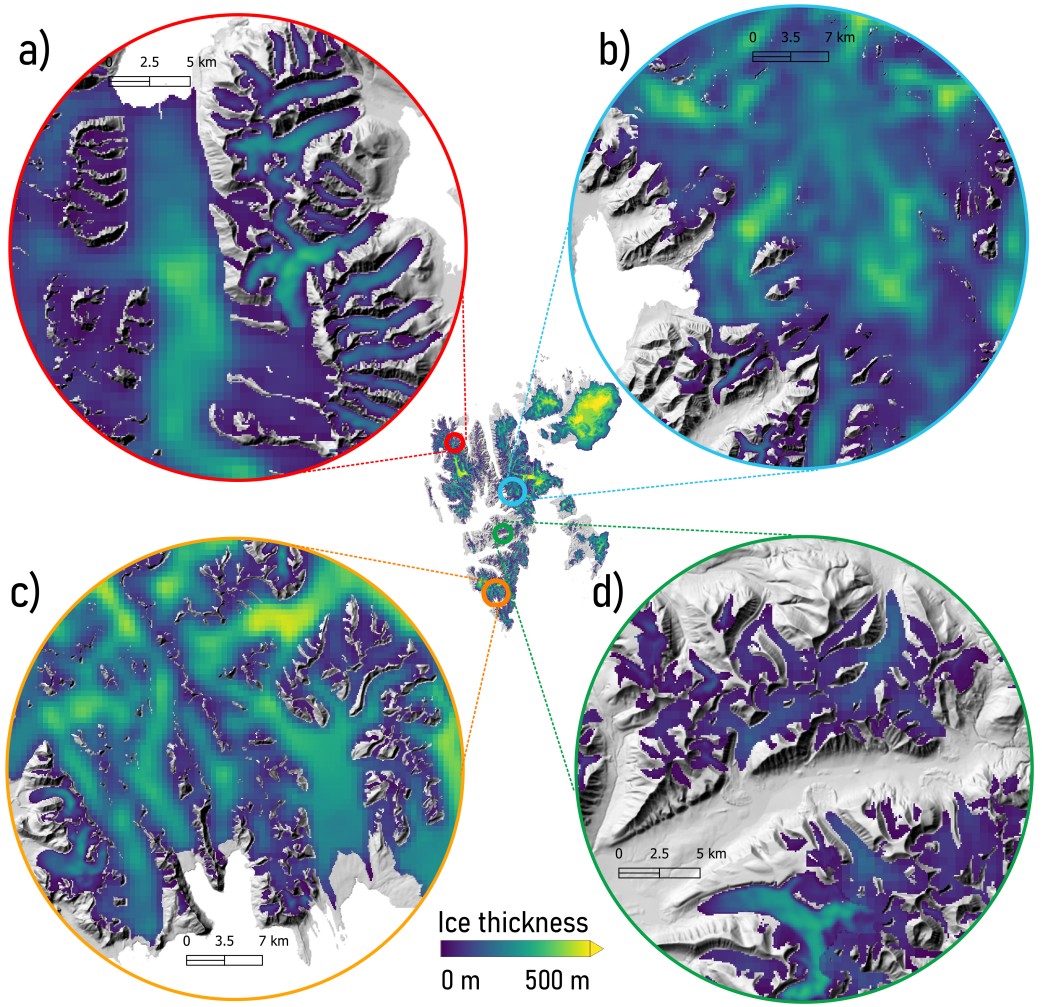

**Figure 4.** Ice thickness in selected regions in northwest (a), central (b,d), and southern Svalbard (c).

Spitsbergen) and Hinlopenbreen (eastern Spitsbergen). Ice thickness for a selection of four regions (Fig. 4) shows how thickness estimates from IGM and PISM are combined; thickness maps for small land-terminating glaciers contain more spatial detail (100-m resolution) than other glaciers (500-m resolution).

A glacier-averaged thickness comparison for tidewater (TW) and land-terminating (LT) glaciers is shown in Figure 5. The median glacier-average thickness is about four times larger for tide-water glaciers (162 m) than for land-terminating glaciers (42 m). These median values are much lower than the Svalbard-wide mean ice thickness (205 m), which results from a skewed size-distribution with a predominantly small and thin glaciers in both glacier categories (LT and TW). Both land-terminating and tidewater glaciers are on average thickest in northeast Svalbard (LT: 55 m, TW: 183 m) and least thick in northwestern Svalbard (LT: 33 m, TW: 114 m). There are 7.5 times more land-terminating glaciers (1,363) than tidewater glaciers (181),

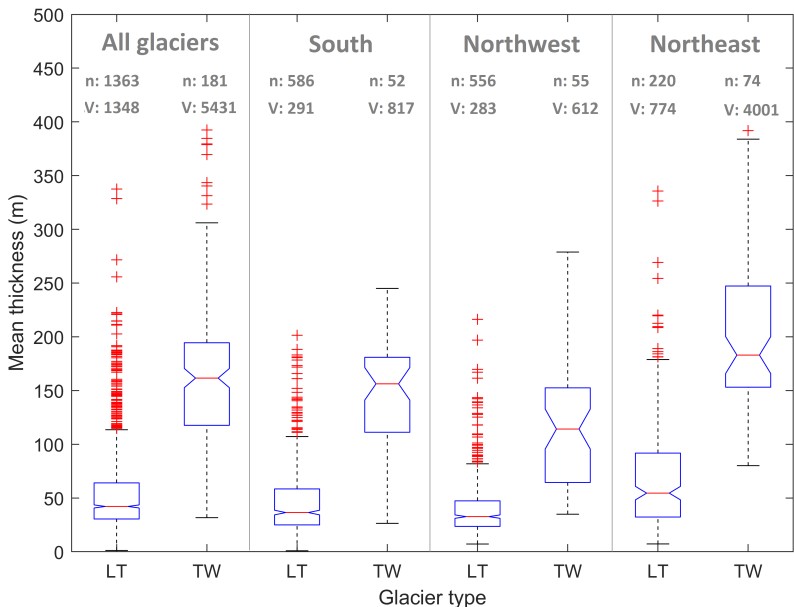

**Figure 5.** Boxplot showing glacier-averaged thickness for land-terminating (LT) and tide-water (TW) glaciers for all glaciers, and split into southern, northwestern and northeastern glaciers. The box-plots for LT and TW glaciers are based on mean thickness values for every glacier. In each category, n is the number of glaciers and V is total glacier volume (in $km^3$). Region boundaries for south, northwest and northeast Svalbard are shown in Fig. 1c.

however, land-terminating glaciers only comprise 20% (1,348 $km^3$) of the total glacier volume. Basin-3 on Austfonna is Svalbard's largest glacier, both in terms of area (1,226 $km^2$) and volume (421 $km^3$). Etonbreen, Austfonna, is the glacier with the largest average thickness (393 m). Primarily due to the small glacier size, no thicknesses could be estimated for a glacier area of 29 $km^2$, equivalent to 0.09% of the total glacier area, and, given their below-average thickness, an even smaller fraction

of the total glacier volume.

The area and volume distributions with elevation for glaciers in southern, northwestern, and northeastern Svalbard (Fig. 6; regions defined in Fig. 1c), show that the volume and area both peak at surface elevations equivalent to (southern Svalbard) or slightly above the equilibrium line altitude (ELA; northwest and northeast Svalbard) in 1957-2018 (Van Pelt et al., 2019). With an expected rise of the ELA (Van Pelt et al., 2021), strongest in southern Svalbard, the relative size of the accumulation zones to

320 the total glacier area (accumulation area ratio) will drop from 43 to 6 % in southern Svalbard, 58 to 27 % in northwestern Svalbard, and 71 to 41 % in northeastern Svalbard from 1957-2018 to 2019-2060. Similarly, the ice volume with a corresponding surface elevation above the ELA will drop from 35 to 4 % in southern Svalbard, 58 to 24 % in northwestern Svalbard, and 77 to 45 % in northeastern Svalbard. The marked drop in southern Svalbard can in part be ascribed to a pronounced narrow peak in hypsometry at low elevations, as previously discussed in Noël et al. (2020) and Van Pelt et al. (2021). Furthermore, it can be

argued that the glacier state, in terms of accumulation area ratio, in northeastern Svalbard in 2019-60 is comparable to the state in southern Svalbard in 1957-2018, i.e. changes in northeastern Svalbard are trailing changes in southern Svalbard by around

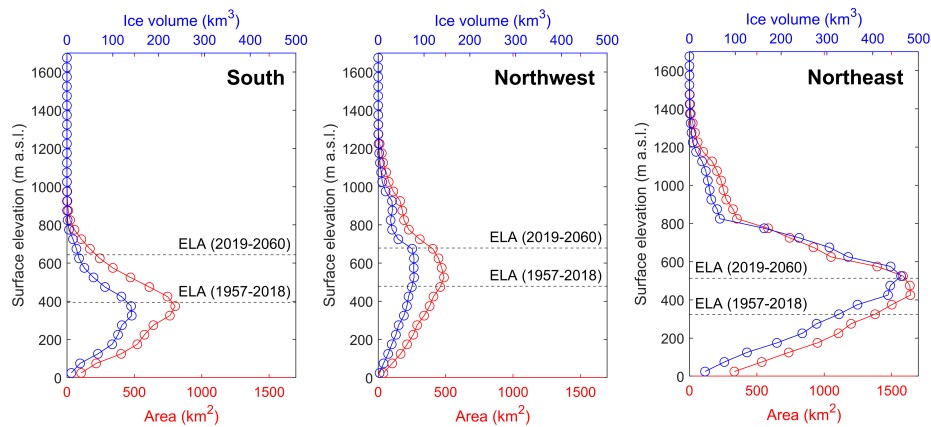

**Figure 6.** Glacier area (red) and volume in 50-m elevation bins in south (left), northwest (middle) and northeast Svalbard (right). ELA values for 1957-2018 and 2019-60 are taken from Van Pelt et al. (2019, 2021). Region boundaries for south, northwest and northeast Svalbard are shown in Fig. 1c.

**Table 2.** Comparison of thickness products against point measurements in GlaThiDa.

| Thickness dataset | R | MAE (m) | RMSE (m) | Bias (m) |
|---|---|---|---|---|
| This study (classes 2 and 3; 500 m) | 0.81 | 57.2 | 75.5 | 0.2 |
| This study (class 1; 100-m) | 0.78 | 37.6 | 49.2 | 0.6 |
| Millan (classes 2 and 3; 500 m) | 0.71 | 81.1 | 107.2 | 23.1 |
| Millan (class 1; 100 m) | 0.76 | 38.0 | 49.1 | -11.4 |

six decades. Finally, it is noteworthy that the above analysis of area and volume responses to ELA changes disregards the amplifying effects of an associated drop in the surface height as glaciers thin. Hence, the presented reductions in accumulation area ratio and volume above the ELA should be regarded as conservative estimates.

## 4.2 Comparison with thickness data & other studies

Since the GlaThiDa thickness data were only used to optimize spatially independent, i.e. global, model parameters, the thickness observation dataset is useful to validate spatial thickness variability. A point-by-point comparison of modeled and observed thickness values is shown in Figure 7. It should be noted that estimated thicknesses are available at two different resolutions (100 m for glaciers in class 1, and 500 m for glaciers in class 2 and 3). It therefore is not feasible to perform a direct comparison for all data at once, as it would involve rescaling (downscaling or averaging) of one of the two datasets to create a dataset with constant spatial resolution; the rescaling itself would affect performance metrics of the rescaled data. Based on the above, we

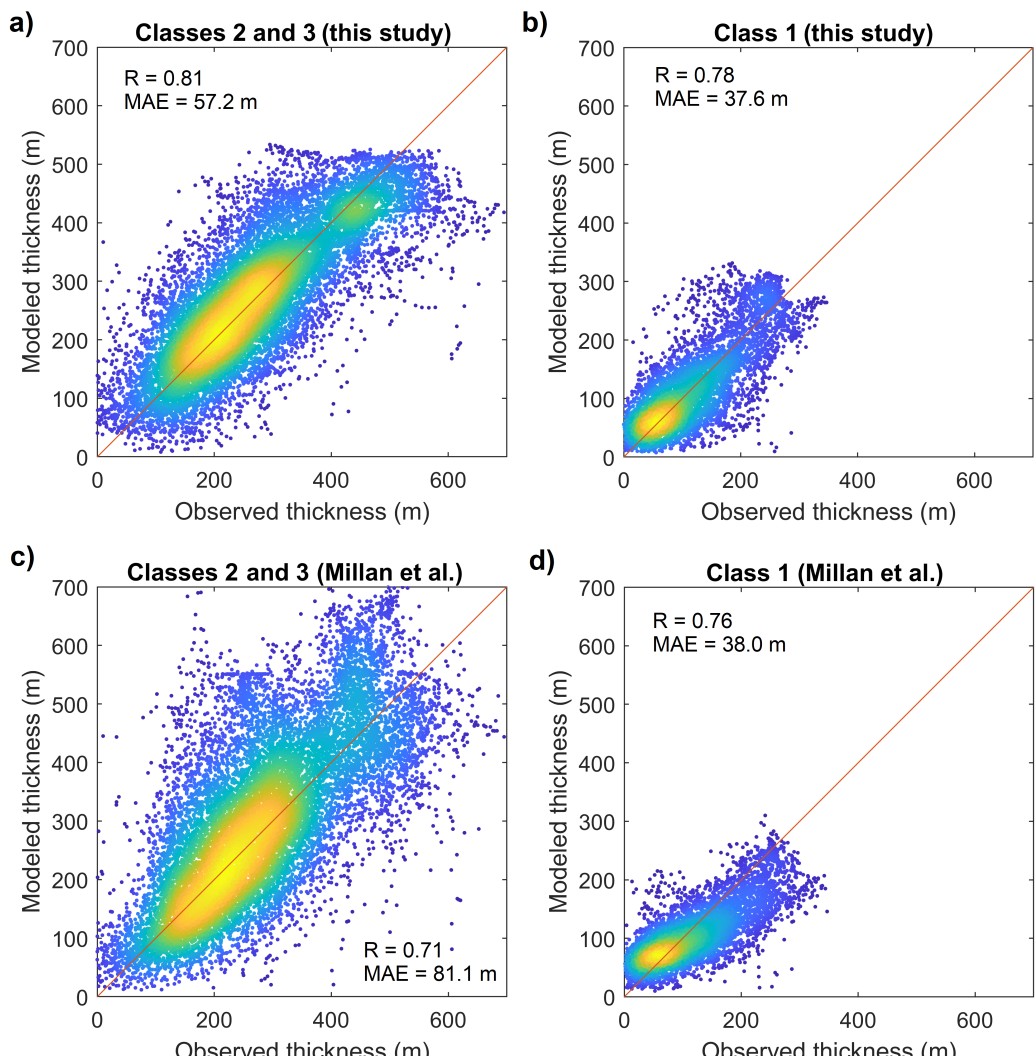

**Figure 7.** Comparison of modeled and observed ice thickness for output from our study (a-b) and Millan et al. (2022) (c-d), and split into data for glaciers in classes 2 and 3 (a and c) and class 1 (b and d). Thickness observations are from the GlaThiDa database (GlaThiDa Consortium, 2020). The comparisons in a and c are based on 500-m resolution output, whereas the comparisons in b and d are based on 100-m resolution output. The dot color represents the density of data points, ranging from dark blue (lowest density) to bright yellow (highest density).

instead perform a comparison of estimated and observed thicknesses at two different resolutions, i.e. at 500 m (glaciers in classes 2 and 3) and 100 m (glaciers in class 1). Observed thicknesses on the 100 and 500 m grids were estimated by averaging all point observation data falling within every 100 or 500 m grid cell respectively.

For all glaciers in classes 2 and 3, we find a mean absolute error of 57.2 m, root-mean-square error of 75.5 m and R-correlation of 0.81. This can be compared with a higher RMSE of 107.2 m and lower $R = 0.71$ by Millan et al. (2022) for the

same glaciers. For all glaciers in class 1 (at 100 m resolution), we find that Millan et al. (2022) produce a similar match with the observations with an MAE of 38.0 m (versus 37.6 m in this study), RMSE of 49.1 m (versus 49.2 m in this study) and $R = 0.76$ (versus $R = 0.78$ in this study). Millan et al. (2022) do experience a considerable negative bias of -11.4 m (versus 0.6 m in this study) for glaciers in class 1 and conversely, a strong positive bias of 23.1 m (0.2 m in this study) for glaciers in classes 2 and 3, suggesting an overestimation of thickness for large glaciers and an underestimation for small glaciers. The scatter plots in Fig. 7a-b reveal that the clouds of points are distributed well around the 1:1 line, suggesting no apparent biases for small or large thicknesses. This is an indication that the degree of smoothness/detail in the bed (height of bed peaks and depth of subglacial troughs) is modeled well, e.g. a too-smooth bed would have resulted in an underestimation of large thicknesses and overestimation of small thicknesses. Similar scatter plots comparing thicknesses by Millan et al. (2022) with observations (Fig. 7c-d) show that the larger errors for glaciers in classes 2 and 3 (Table 2) are a result of a general larger spread in the Millan et al. (2022) dataset, primarily for large thicknesses. For small glaciers (class 1) Millan et al. (2022) show an underestimation of large thicknesses and an overestimation of small thicknesses, indicating that the Millan et al. (2022) thickness product is smoother than reality.

It should be noted that in the case that PISM was used for the glaciers currently modeled with IGM (class 1), MAE would increase to 42.7 m (IGM: 38.0 m), RMSE to 54.1 m (IGM: 50.1 m) and R would drop to 0.71 (IGM: 0.77). For this comparison, PISM results on the 500-m resolution grid were reprojected to the 100-m resolution IGM grid using nearest-neighbor interpolation (running PISM at 100-m resolution is too computationally costly and results in violation of the shallowness assumptions). The above confirms that the use of IGM for small glaciers leads to better agreement with thickness measurements. One reason may be the higher-order physics behind IGM, which helps to resolve small-scale ice flow and bed features better than with a model like PISM which is based on shallowness assumptions (i.e. small depth-to-width ratios are less likely to apply to glaciers in class 1). The superior performance of IGM for small land-terminating glaciers was the main reason to use two different ice flow models for glacier classes 1 and 2. IGM is under constant development, and to date no extensive tests have been performed yet on grounded tide-water glaciers. Using IGM and the same input datasets and model assumptions as with PISM we performed first tests on a selection of large (tidewater) glaciers in Svalbard showing slightly worse performance (more details in Response to Reviewer 1). This may lie in the machine-learning character of IGM, which can only approximate the results of conventional ice flow models that directly solve the stress equations. It is also worth noting that IGM experiences a loss of accuracy with increasing domain size (Jouvet and Cordonnier, 2023), further underscoring that IGM does not generate a replica of regular higher-order model results.

A spatial comparison of our thickness map with previous maps presented in Millan et al. (2022) and Fürst et al. (2018b) is shown in Figure 8. Millan et al. (2022) found a similar volume (6,855 km$^3$) and average thickness (207 m), while Fürst et al. (2018b) (version 1.1) found higher volume (7,213 km$^3$) and mean thickness (220 m) estimates. It should be noted that, in contrast to Millan et al. (2022) and this study, Fürst et al. (2018b) locally calibrated their method against point thickness observations, implying that thickness observations are imprinted in the thickness product. Based on this, we excluded Fürst et al. (2018b) from the thickness comparison in Table 2. In general, our study shows more similarities in terms of spatial distribution with Fürst et al. (2018b) than with Millan et al. (2022), as shown by the lower overall deviations from our thickness map (Figs.

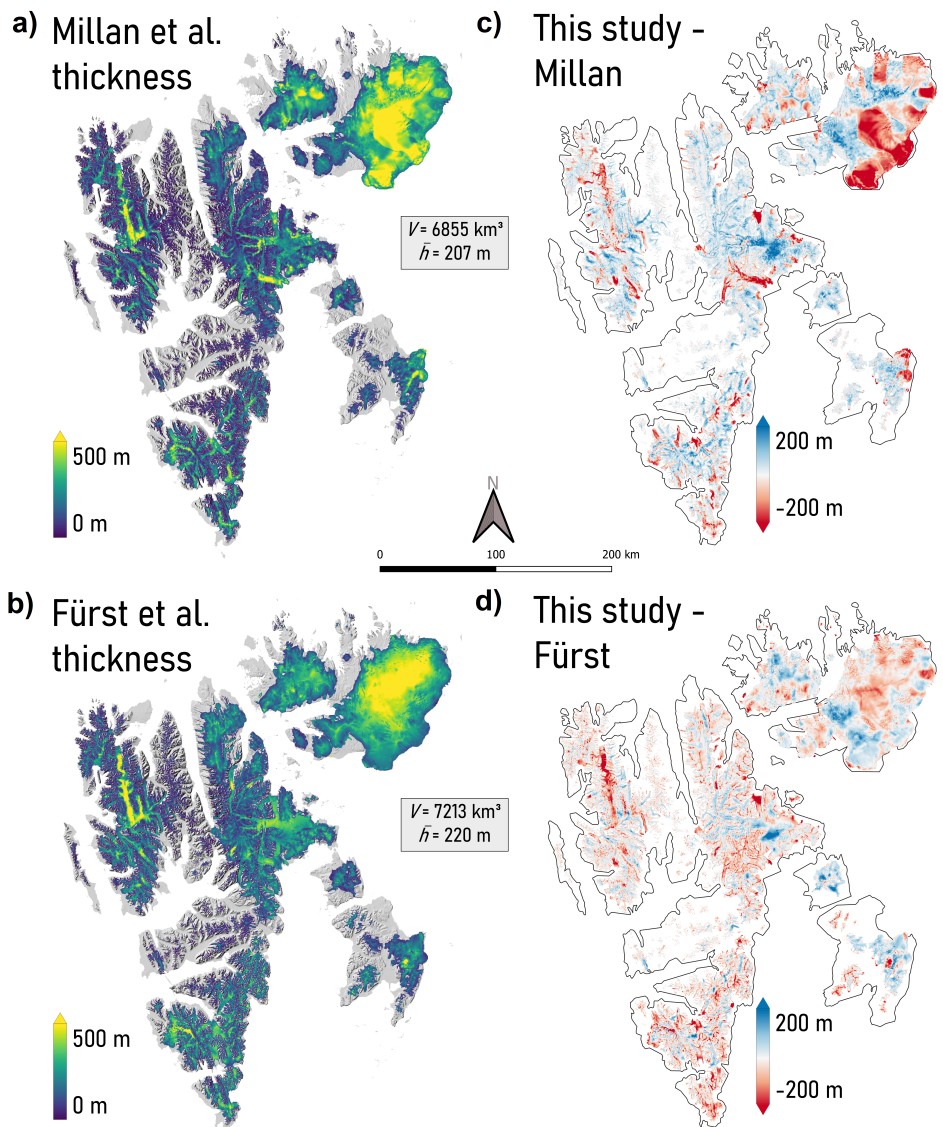

**Figure 8.** Previous ice thickness maps by Millan et al. (2022) (a) and Fürst et al. (2018a) (version 1.1) (b), and the corresponding differences with our results (c-d).

8c and d). The better agreement of our study with Fürst et al. (2018b) than with Millan et al. (2022) may in part reflect the better agreement of our product with the thickness data (which are integrated in the Fürst et al. (2018b) thickness map). For the large Austfonna ice cap, our study and Fürst et al. (2018b) are in better agreement than our study and Millan et al. (2022); most notably our study and Fürst et al. (2018b) experience less pronounced jumps near ice divides.

The inverse method in Millan et al. (2022) relies on ice velocities and inversion of the SIA, with a parameterized description of sliding, to estimate thickness. The overestimation of ice thickness for large glaciers in Millan et al. (2022) (Table 2), and most

prominently for surging glaciers, e.g. Basin-3, Tunabreen, Negribreen and Storebreen (Fig. 8) could result from inappropriate physics to describe the highly dynamic and complex flow. The same argument, in addition to mismatches in the time stamps of input datasets, has led us to use the simpler perfect-plasticity method for surging glaciers in this study. Regarding the comparison with Fürst et al. (2018b) we note that Fig. 8 compares our product against version 1.1 of the Fürst et al. (2018b) dataset which differs considerably (e.g. 20% higher volume) from version 1.0 that was described and presented in the paper. It is noteworthy though that the Fürst et al. products can be seen as an "interpolation method" as the observations are imprinted in the map and mass conservation and viscosity tuning are applied to generated thickness in between observations. Our study is less informed by the observations (only to constrain global parameters) which we argue leads to a map that may be more consistent in space (in terms of spatial detail/roughness and uncertainty) and has the advantage that it can be used as a numerically stable spin up state for prognostic modeling. This currently however only applies to glaciers in classes 1 and 2, for which iterative inverse methods were used. In case also glaciers in class 3 are to be included in a prognostic run, we would suggest to instead use PISM also for these glaciers to allow for spinup and transient forward modelling (as for glaciers in class 2). This inevitably does introduce larger uncertainty in the basal topography and initial ice thickness.

## 4.3 Uncertainties

By applying dedicated inverse methods and model physics for different glacier types, using state-of-the-art remote sensing and model input datasets, and calibrating against thickness observations, we limit uncertainties in the final thickness and bed maps. Arguably, using different ice flow models, spatial resolution, and individual parameter calibration per glacier class, causes some consistency between the methods to be lost. However, advantageously we achieve a lower misfit with thickness observations by treating glacier types separately. More specifically, the superior performance of IGM for glaciers in class 1, as well as the improved results with PISM for glaciers in class 2, were the main reasons to use two different ice flow models for these classes. Regarding the use of different spatial resolutions, we emphasize that there is a limit to the degree of detail in the bed that can be recovered from inversion, which scales with the ice thickness (Gudmundsson and Raymond, 2008). Hence, smaller-scale bed details can theoretically be recovered for smaller (thinner) glaciers than for larger (thicker) glaciers. This supports the use of different spatial resolutions for different glacier sizes. In summary, our modelling choices led to more detailed bed and thickness maps that are in closer agreement with observations, yet at the expense of some coherency.

In the hypothetical case of perfectly accurate ice flow physics, and flawless and synchronous input datasets (climatic mass balance, surface height, surface height change and surface velocity), an error-free bed map (except for fine-scale topography) can be generated with iterative updates of basal boundary conditions (bed height and friction) in an ice flow model. Although this is fictitious, it does give directions for future improvement of inverse estimation of basal conditions, which among others demands a better description of ice flow physics, and higher quality and synchronous input and validation datasets. For a more extensive discussion on thickness error sources, e.g. from inaccurate model physics, inverse model parameters, and noisy input datasets, we refer to Frank et al. (2023) and Frank and Van Pelt (2024).

The validation of local ice thicknesses against available observations (Sect. 4.2) gives a direct estimate of the uncertainty of bed heights and thicknesses for these locations. Instead, the total volume uncertainty cannot be directly quantified and is

here based on the assumption that it is the sum of errors resulting from uncertainty in glacier extent (extracted from the RGI database) and modeled mean thickness. The large and well-distributed thickness observations dataset available for Svalbard used for model calibration, including data from 169 glaciers, helped to reduce the Svalbard-wide volume uncertainty (estimated at 3.5%). Whereas the RMSE of Svalbard mean glacier thickness is only 3.5 m as a result of averaging and calibration, the local (point) thickness error is considerably larger (49.2 m for class 1 and 75.5 m for class 2 and 3, Tab. 2). The volume uncertainty may be underestimated in case the uncertainty of glacier extent in the RGI outlines for Svalbard is larger than the 1-2% that Nuth et al. (2013) estimated. Furthermore, systematic biases in thickness observations (e.g. instrumental or data processing errors such as radar travel time to thickness conversions) may create additional volume uncertainty, although there are no indications for this.

Given the different (average) timings of input datasets, it is hard to set a date for the bed and thickness maps. A rough best estimate would be 2010-2015, which is the median for key input datasets of surface height, surface height change, ice velocity, climatic mass balance, and glacier outlines. Ice thickness observations in GlaThiDa have been collected from 1983 to 2016, and represent a mean date ($\sim$year 2009-2010) that is three years earlier than the representative date of the model output. With previously estimated thinning in Svalbard of $\sim$-0.35 m per year in 1936-2010 (Geyman et al., 2022), i.e. 0.17% relative volume loss per year, the real volume in 2010-2015 may have been $\sim$35 km$^3$ smaller than we modeled. Similarly, a retreat of glaciers of -39 km$^2$ per year (1936-2010; Geyman et al., 2022), or a relative area loss of 0.12% per year, implies an additional potential volume loss of -39 km$^3$ between the mean collection date of glacier outlines (2007-2008) and the reference time for our thickness map. These volume bias estimates should be regarded as rough estimates, e.g. as the actual rates of area and thickness change may have differed from the used 1936-2010 averages. The different timing of input datasets complicates the inversion of thickness for glaciers that experience rapid geometric and dynamic changes. This particularly applies to surging glaciers, where the application of iterative inverse methods would introduce excessive errors primarily due to timing mismatches between surface height, surface height change, and velocity datasets. In case such timing mismatches can be reduced, we would recommend the use of iterative inverse methods also for surging glaciers in future experiments.

## 5  Conclusions

We present a new bed height and thickness map for all glaciers in Svalbard, generated using a combination of three inverse methods. Combining the methods allows us to simulate small land-terminating glaciers with high spatial resolution (100-m) using the deep-learning model IGM, whereas thickness inversion for large tidewater and land-terminating glaciers benefits from a SIA+SSA approach in PISM to describe sliding motion. Input data uncertainty for actively surging glaciers led us to use a simple perfect-plasticity-based method for those glaciers. Comparison of thicknesses with observations reveals good agreement with point observations for glaciers of different types. Particularly, for large and tidewater glaciers we find improved estimates of ice thickness compared to a previous study by Millan et al. (2022). We find that Svalbard's glaciers, excluding Kvitøya, have a volume of 6,800$\pm$238 km$^3$ (16.3$\pm$0.6 mm sea-level equivalent) and a mean thickness of 205$\pm$7 m, which is

in between recent estimates of 5,963 km$^3$ or 182 m (Fürst et al., 2018b), 7,213 km$^3$ or 220 m (Fürst et al., 2018a), and 6,855 km$^3$ or 207 m (Millan et al., 2022), generated using entirely independent methodologies.

The bed and thickness datasets are made available in open-access databases and may find further applications within glaciology and other fields (e.g. in studies of runoff and impacts on fjord processes). A benefit of thickness maps produced with iterative inverse methods, i.e. for all not actively surging glaciers, is that they simultaneously provide initial conditions for future simulation of the same set of glaciers. However, this does require the use of the same ice flow model, setup, and temporal consistency of input datasets.

*Code and data availability.*  The bed and thickness datasets, presented in Fig. 3, together with the mask shown in Fig. 1c, are uploaded as Geotiff-files to the following repository: https://doi.org/10.5281/zenodo.11239460. The source code of the Parallel Ice Sheet Model can be accessed at https://www.pism.io/. The Instructed Glacier Model is available at https://github.com/jouvetg/igm.

*Author contributions.*  Both authors (WvP and TF) contributed equally to the study design, the modelling experiments, data analysis and manuscript writing.

*Competing interests.*  There are no competing interests.

*Acknowledgements.*  WvP acknowledges funding from a career grant by the Swedish National Space Agency (2018-C; project nr. 189/18) and a starting grant by the Swedish Research Council (project nr. 2020-04319). Development of PISM is supported by NASA grants 20-CRYO2020-0052 and 80NSSC22K0274 and NSF grant OAC-2118285.

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
