# Peer review of "New glacier thickness and bed topography maps for Svalbard"

_EGUsphere, 2024_

## Referee Comment (RC1)

Summary and comments on the manuscript
egusphere-2024-1525 entitled

**A new glacier thickness and bed map for Svalbard**

presented on 03.06.2024
by

Ward Van Pelt and Thomas Frank

**SUMMARY**

The title directly sets the stage for the overarching objective,
i.e., to forward a new map of glacier ice thickness for Svalbard.
For this purpose, the authors employ a state-of-the-art method that
builds on surface observations of geometry and velocity as well
as model estimates of surface mass balance. Calibration target
is the abundant record of thickness measurements. For the actual
mapping, the authors distinguish three glacier types using specific
treatment for active surges. Apart from the actual thickness distribution,
the authors ultimately report a total ice volume of $6800\pm238$ km$^3$,
which is within the range of previous estimates.

Admittedly I am very excited about this new mapping effort on Svalbard.
The reason is that all previous attempts have their specific weaknesses.
The manuscript is well written and strikes with clearness and high-quality
illustrations. Altogether it is very easy to follow. When I read
the manuscript, one question got stuck in my head. Should this
new map replace previous efforts (new standard) and if yes, what
are the key arguments for the quality increase. In my view, the
manuscript fails to explain this. Apart from that, I see some methodological
aspects requiring further justification or adaptation (e.g., glacier
classes, calibration, multi-model approach, uncertainty). Overall,
I remain very positive about this manuscript and I recommend that
the editor should continue to considered it for publication in *The
Cryosphere* after my main concerns below have been alleviated.

**MAJOR COMMENTS**

**GLACIER CATEGORIES**
The criteria for defining several glacier classes are well presented.
You distinguish in terms of area (threshold 100km2), termination

type (land/marine) as well as observed surge activity in 2017/18.
Yet later in the manuscript (L116-117), you diffuse these categories
again by joining well-connected ice geometries and with it combining
different classes. So a land terminating glacier smaller than 100
km2 that is connected to a larger ice-body will be treated differently
than its stand-alone homologue. The same is true for surge-type
glaciers. All surge-type glaciers are embedded in larger icefields
(Fig.1c). Does actually any glacier remain in class 3? If yes
and they fall into an icefield, what is done at the its internal
boundary? In summary, the classification appears somewhat confusing
to me. I am sure you find a more consistent strategy.

**METHODOLOGICAL MIX**
I understand that you sell the glacier classification, and with
it the specific methods per class, as a strength of your approach.
The Parallel Ice Sheet Model (PISM) is used for the larger glacier
compounds (class 2), the Instructed Glacier Model (IGM) for the
smaller land-terminating glaciers (class 1) whereas surge type glaciers
(class 3) are treated with the perfect plasticity approach (requiring
minimum input). While it is clear that surge type glaciers need
to be treated differently, I do not get my head around it, why two
approaches are necessary for the other two classes (1 and 2). You
argue that IGM comprises higher-order dynamics necessary for the
smaller glaciers. Yet, higher-order dynamics would also be preferential
for the larger ice bodies. Some of these are marine terminating
and show significant flow speeds near the ice front. I do not see
the advantage of using two models especially as the glacier class
definition is a bit diffuse (see above). Last but not least, you
need to say something about the IGM capabilities of inferring ice
thickness itself without integration into a transient assimilation.
Please add this to the discussion.

**PROGRESS & DISCUSSION**
As much as I like it that a new thickness map of the Svalbard ice
cover is presented, I wonder about the improvements with respect
to existing products. You can either show that the quality of the
input data is higher or that your method is more sound. Alternatively,
you simply show some performance measures by comparing the different
thickness products and leave the decision to the reader. I am not
sure what is the best way forward in your case. The most practical
is to extend your discussion by additional analysis. I suggest
that you simply add previous results to some existing figures (Figs.
5, 6 and 7). This extension will help readers to better assess
your new thickness product. In case you find further arguments

to promote your map product, please stress this prominently throughout the text and certainly in the abstract. One last thought that got stuck in my head: the two previous estimates (Fürst & Millan) seem to differ in volume and spatial distribution. Your map reproduces the volume of one approach and the pattern of the other. Why is that? Could this give an argument?

**INPUT DATA**
I would appreciate a little regional overview of all input data in Sect. 2. Many of these data sets have global coverage and it is difficult to assess their quality on Svalbard. You should comment on that briefly. Admittedly, you have Fig. 1 but nothing is said about uncertainties/quality. Possibly some of them are better suited than what was used for previous Svalbard maps.

**CALIBRATION** If I understand your manuscript correctly, you calibrate several parameters (spatially uniform) for both the IGM and the PISM inversion. IGM and PISM only differ in the a-priori choice of the rate factor and the sliding coefficient as well as in the SMB correction strategy. Actual calibration parameters are $\beta$, $\Theta$, $\tau$ and $\alpha_{min}$. These mostly relate to the initial guess for ice thickness and the iterative inversion procedure. I cannot understand why you would need different initial guesses for these two models with $\tau_{PISM} = 0.52kPa$ vs. $\tau_{IGM} = 100kPa$ and $\alpha_{PISM} = 0.014$ vs. $\alpha_{IGM} = 0.04$. Isn't the perfect plasticity approach only calibrated once? Second, I am puzzled why the iterative inversion method requires so different values: $\beta_{PISM} = 0.25$ vs. $\beta_{IGM} = 1$ and $\Theta_{PISM} = 0.4$ vs. $\Theta_{IGM} = 0.15$ (also no friction update in IGM). Can you please explain? These calibration differences cast doubts on keeping a consistent map product while applying two glacier-system models.

**MINOR COMMENTS**

**L91** Please clarify why you chose the perfect plasticity approach for surge-type glaciers. In my view, your main reason is the temporal consistency of the input data. The method by itself is not more adequate for such glaciers. I am not sure if all readers immediately get this.

**L107-108** It is not very clear why you limit your surge-type class to the years 2017-2018. Koch et al. (2023) present surging glaciers for a longer time period. Why did you refine your selection? Your DEM dates back to 2010. So many other surges are imprinted. Do

you see a problem from that even using the perfect plasticity.

**L126** From my understanding, the term yield stress relates to when deformation becomes possible whereas the term sliding law normally relates basal velocities to general basal stress conditions. Please check this terminology and be specific about any assumptions.

**L136** In the in-line equation here, you relate two stresses to each other, namely $\tau_d$ and $\tau_c$. Velocity is the scaling factor. In this definition, $\tau_d$ and $\tau_c$ cannot both be stresses. I would consider one of them a basal friction coefficient. I therefore would not use the symbol $\tau$.

**L166** Here, you present the first calibrated parameters. $\Theta$ represents the strength of the surface update. I am very surprised by its magnitude of 40%. This means that if you have to adjust the thickness at one location, 40% of this change will be imprinted in the surface. This is a lot. I wonder how much your modelled surface then deviates from the observed one after convergence. Please clarify.

**L205 & L226** I am happy to see such small threshold values here. Please give them in degrees.

**L219** time periods --> time stamps

**L254-256** Your uncertainty estimate of the mean thickness of 205 m is ±7 m . As you say, this is about ±3.5%. Measurement errors of thickness observations typically exceed 10-20% and therefore strongly challenge your estimate. These measurement errors are not considered in your uncertainty analysis. Moreover, I do not get my head around your argument (L255) that the standard deviation of modelled vs. observed thickness values must be divided again by the number of glaciers with observations (i.e., $\sqrt{169}$) Please explain better or remove this division.

**L270** For the land- and marine terminating glaciers, you report mean thickness values of 42 m and 162 m, respectively. How can you reconcile this value with the archipelago-wide average of 205 m. I certainly miss something here ;o)

FIGURES
**Fig. 1** I like this figure very much. In panel (c) you indicate the locations of all surge-type glaciers. They all belong to larger ice-fields. So are they now modelled with PISM or the perfect plasticity approach. In the latter case, how do you ensure that you do not get internal fringe lines in the thickness field.

**Fig. 2** Could you add the reference run with $M_{corr} = 0.4$ m w.e. $yr^{-1}$

and $\Theta = 0.4$.

**Fig. 4** In panel (a) and (b), the ice thickness decreases towards the calving fronts of marine-terminating glaciers. Why is that? On Hansbreen, you could directly compare to a dense survey grid included in GlaThiDa.

**Fig. 4 & 5** Please extend these figures by values from the existing thickness maps. It might also help in evaluating & promoting your results.

**Fig. 7** How does this distribution look like for Millan? It would be good to add for reference. It might also help in evaluating & promoting your results.

**Fig. 8** Please use the updated thickness map (v1.1) from Fürst et al. (2018) at data.npolar.no.

TABLES
**Table 1** Please indicate the version of GlaThiDa.

**Table 2** Please use the numbering you introduced for the glacier classes (1-3).

CODE AVAILABILITY
Is the inversion method available via an open repository?

---

## Referee Comment (RC2)

**Review of van Pelt and Frank (2024): A new glacier thickness and bed map for Svalbard**

**Summary**
This paper presents a new ice thickness and bed map for Svalbard based on a method previously developed by the authors. Here, they use three different methods for small, land-terminating glaciers (IGM), larger and tidewater glaciers (PISM), and surging glaciers (perfect plasticity) to derive their results, having performed a considerable amount of calibration and processing to reduce errors and ensure agreement across their results (no big jumps in ice thickness at ice divides). They then compare their results to other recent bed-thickness datasets for Svalbard, showing that their work sits within the expected range, but substantially reduces errors and bias across the board.

I think this is an innovative paper that attempts to leverage recent developments to obtain the best-possible results. However, I have a few major concerns, as well as several minor ones before I would be happy to recommend the paper for publication. I wonder whether the use of three different methods, particularly when it is not clear to me the rationale for using two different ice-flow models, has not overcomplicated the paper and sacrificed internal consistency – normally an advantage of these large-scale datasets – for an unclear gain in accuracy. The authors establish using PISM where they use IGM would lead to a worse outcome, but not whether the reverse is true, which seems to me a major oversight that makes it difficult to see what advantage using PISM and complexifying the method really brings. I am also very unclear as to how the authors set up their model domains and how they dealt with the resulting boundary conditions, which makes it difficult for me to assess the quality of their modelling. Overall, I therefore think major revisions are required: it may just be a question of adding/clarifying some information that is not obvious as the paper is currently written, which I hope is the case, as I think the end outcome and method are very interesting!

I should also note that I read the paper and wrote the review before I read Reviewer 1's comments. The fact that both of us largely raise the same issues is therefore not a result of groupthink.

Line and page numbers refer to those in the manuscript.

**Major Points**
- Consistency: this is perhaps something of a more philosophical point, but a major advantage of these kinds of large-scale studies is that (usually) they apply the same processing steps to a wide area so that, even if one doesn't believe the absolute numbers very much, one can be confident that the results are internally self-consistent. Here, the use of three (very) different methods means this cannot be taken for granted in the same way. I think the authors have done a lot of work to try to overcome this, particularly with interpolating results onto different grids, but I wonder if this was the right approach.
- Choice of methods: I do not think the authors really provide a clear justification for why they chose to use PISM for the larger (and tidewater) glaciers, as opposed to IGM. I take the point that PISM in SIA+SSA mode does a good job for larger and tidewater glaciers, but IGM is a higher-order model that would also do a good job here. Usually, this choice would be justified by saying that higher-order models are too computationally expensive to make them practical at this scale, but IGM has been explicitly designed to run fast and overcome that objection. So what motivated the choice? Because, if the authors had used IGM for all the larger and tidewater glaciers, that would have allowed them to achieve a much larger degree of consistency in the results and avoided them a considerable degree of work at the calibration, method and post-processing stage, it seems to me (of course, they would have had to find a strategy to vary the sliding parameter in IGM, but that feels to me an easier thing to do than have two separate methods working on different assumptions at different resolution)
- Boundary conditions and model domains: I am unclear how the authors defined their model domains. I assume, given they use RGI6.0, that they take each RGI outline and invert it individually? Or do they take all contiguous RGI6.0 outlines and invert them as one entity? In the former case, how do they then deal with ice-ice boundaries, where two different RGI entities are in contact? In both cases, what boundary condition is imposed at the front of tidewater glaciers? As both the PISM

and IGM methods involve small forward timesteps, these issues need to be considered. At the very least, a few lines in the discussion about how not considering these likely introduces some local inaccuracies need to be added.

- Discussion: Ultimately, I think this comes back to my point on the choice of methods above, but I don't find that the discussion does a very good job of highlighting what this study brings to the table and why people should use the bed calculated in this study as opposed to those from other studies. The authors provide plenty of description for how their results compare to other datasets, but mostly do not analyse why these differences occur, making it hard for readers to assess which product is better for their particular application. The fact that it is also unclear as to why the authors made particular methodological choices (see my comment above) then further muddies the waters here. The authors do show that they substantially reduce the error on larger glaciers compared to previous studies and the bias across all glaciers (Table 2), which I would argue is the main selling point of their results in the current formulation of the paper, but this gets a bit lost in the discussion and no mention of it is made in the abstract (there is a partial reference in the conclusion, but only to the error on larger glaciers), making it very easy for readers to lose sight of it completely.

**Minor Points**
- p.1, l.10: I might venture to say that a mean ice thickness at the scale of the whole Svalbard archipelago isn't that useful or meaningful a number to include in the abstract (at least, as a headline figure for the paper, it seems some way down the list of things that readers would want to know)? The total volume, yes, but I would suggest maybe converting that into an SLR equivalent for the second number, or reporting the maximum ice thickness, which is something that makes a bit more sense at that scale. Or, possibly even more useful, say something about how the volume estimate presented here compares to other studies' estimates.
- p.2, l.42: Reference formatting for Farinotti et al.
- Table 1: Why use the 20 m NPI DEM when it has to be downscaled to 100 or 500 m immediately? Wouldn't the COP90 DEM have been a better choice to fit with the modelling resolutions and also sit more in the middle of the range of most of the other data (2010-2019ish)? Also, the RGI6.0 outlines for Svalbard have dates of 2000-2010, so please update the table to reflect that.
- Figure 1: I can see already that, on Austfonna, there are two methods being used to generate the results, despite the assertion in lines 116-117 that all the glaciers connected to larger ice caps are modelled using PISM (i.e. the same method). Can the authors confirm whether the figure or the text is correct here? If the figure is right, how are they dealing with the jumps in thickness at the ice divides on Austfonna?
- p.7, l.169: 'with a'
- p.9, l.200-204: Can the authors comment as to how far using uniform parameter values might introduce some error into the results?
- p.9, l.214: I confess I'm not entirely clear on why the thickness field produced by IGM would have gaps in it that need interpolating?
- p.10, l.231-232: This seems to me quite a substantial upsampling of the majority of the dataset that might introduce a considerable number of artefacts. Would not downsampling the 100 m proportion of the dataset to 500 m have been the more conservative choice?
- p.10, l.245-257: Yes, but are there observed glaciers in each of the three categories, such that all three types of inversion are bias-free? More generally, with three different inversion methods, would there not need to be three separate estimates of sigma H bar, one for each category? Because a mean error of 3.5 m on ice thickness across the whole of Svalbard seems a little too good to be true. The observations themselves would have bigger errors than that!
- p.16, l.314-319: Have the authors performed the same comparison in the other direction? As in, what happens if IGM is used to model the larger glaciers where PISM was the preferred method? Otherwise, I think it's difficult to say that the combination of the two methods is superior to either alone. The approximations in PISM may be suitable on the larger glaciers, but it doesn't follow that that means they're more suitable than using a higher-order model
- p.16, l.321: OK, yes, 6855 is higher than 6800 and 207 is higher than 205, but I'm not sure that it's really a meaningful difference, especially when both those numbers are well within this study's own error bars. Consider rephrasing this to make it clearer that this study's integrated volume and mean thickness results are not significantly different to those from Millan et al.

- p.16, l.320-333: Could the authors provide some more analysis of why these differences exist? They posit sensible reasons for why Millan et al. likely overestimate ice thickness on larger glaciers, but I think it would make the paper much more useful for the community if they can suggest some reasons for the other differences (spatial distribution more similar to Fürst, thicker ice at lower elevations than Fürst, less pronounced jumps at ice divides than Millan, etc.), as it would help people work out which is the best bed product for them to use for their particular application
- p.18, l.384: This is only true provided other people use the exact same set of final modelled bed, velocity, surface, etc. as used in this study as their initial conditions. If someone took the bed from this study and then used, say, the COP90 surface DEM and ITSLive velocities to initialise their model, they would not have a harmonious set of initial conditions. Please rephrase this to make it more clear.

---

## Author Comment (AC1)

**Response to RC1:**

Referee comments in **red**
Author comments in **green**

The title directly sets the stage for the overarching objective, i.e., to forward a new map of glacier ice thickness for Svalbard. For this purpose, the authors employ a state-of-the-art method that builds on surface observations of geometry and velocity as well as model estimates of surface mass balance. Calibration target is the abundant record of thickness measurements. For the actual mapping, the authors distinguish three glacier types using specific treatment for active surges. Apart from the actual thickness distribution, the authors ultimately report a total ice volume of 6 800±238 km3, which is within the range of previous estimates. Admittedly I am very excited about this new mapping effort on Svalbard. The reason is that all previous attempts have their specific weaknesses. The manuscript is well written and strikes with clearness and high-quality illustrations. Altogether it is very easy to follow. When I read the manuscript, one question got stuck in my head. Should this new map replace previous efforts (new standard) and if yes, what are the key arguments for the quality increase. In my view, the manuscript fails to explain this. Apart from that, I see some methodological aspects requiring further justification or adaptation (e.g., glacier classes, calibration, multi-model approach, uncertainty). Overall, I remain very positive about this manuscript and I recommend that the editor should continue to considered it for publication in The Cryosphere after my main concerns below have been alleviated.

We thank the reviewer for the constructive and detailed feedback, which have helped us a lot to improve the manuscript!

MAJOR COMMENTS
GLACIER CATEGORIES
The criteria for defining several glacier classes are well presented. You distinguish in terms of area (threshold 100km2), termination type (land/marine) as well as observed surge activity in 2017/18. Yet later in the manuscript (L116-117), you diffuse these categories again by joining well-connected ice geometries and with it combining different classes. So a land terminating glacier smaller than 100 km2 that is connected to a larger ice-body will be treated differently than its stand-alone homologue. The same is true for surge-type glaciers. All surge-type glaciers are embedded in larger icefields (Fig.1c). Does actually any glacier remain in class 3? If yes and they fall into an icefield, what is done at the its internal boundary? In summary, the classification appears somewhat confusing to me. I am sure you find a more consistent strategy.

Thanks for bringing this up. It was indeed confusing the way it was described in the original manuscript. We still stand behind our original division into the three glacier classes, but we realize the statement about the large ice systems (L116-117) was incorrect / incomplete. Non-surging small glaciers (<100 km2) and large glaciers (>100 km2) that are part of large ice systems (ice caps) are all modeled with PISM to avoid jumps at ice divides. The same however does not apply to actively surging glaciers which are not modeled with PISM but rather with the perfect plasticity assumption, even when they are part of a large ice system. To avoid thickness jumps between surging and non-surging glaciers the simulations with

PISM included also the surging glaciers in the model grid, but held the surface height and thickness fixed for those glaciers while performing the inversion for the non-surging glaciers. This allowed for mass fluxes through the ice divides between surging and non-surging glaciers and avoided thickness jumps within the large ice systems.

We have rephrased L116-117 as follows: "*One nuance to the three groups above is that all (small) glaciers in group 1) that are part of / connected to larger ice caps are modeled with PISM. This is to avoid thickness jumps at the ice divides. Furthermore, to avoid thickness jumps within ice caps between PISM-modeled and surging glaciers, experiments with PISM also include the surging glaciers as static entities with thicknesses based on the perfect-plasticity assumption.*"

METHODOLOGICAL MIX
I understand that you sell the glacier classification, and with it the specific methods per class, as a strength of your approach. The Parallel Ice Sheet Model (PISM) is used for the larger glacier compounds (class 2), the Instructed Glacier Model (IGM) for the smaller land-terminating glaciers (class 1) whereas surge type glaciers (class 3) are treated with the perfect plasticity approach (requiring minimum input). While it is clear that surge type glaciers need to be treated differently, I do not get my head around it, why two approaches are necessary for the other two classes (1 and 2). You argue that IGM comprises higher-order dynamics necessary for the smaller glaciers. Yet, higher-order dynamics would also be preferential for the larger ice bodies. Some of these are marine terminating and show significant flow speeds near the ice front. I do not see the advantage of using two models especially as the glacier class definition is a bit diffuse (see above).

We agree with the reviewer that ideally all glaciers should be modeled with the best possible physics. Following that logic it would have made sense to model also the large tidewater glaciers using IGM. There are however several reasons for us to choose PISM instead for the large glaciers. PISM is an ice flow model that has previously been successfully applied to model glaciers and ice caps that exhibit both sliding and non-sliding flow (by combining the shallow shelf and shallow ice approximations) and calving. IGM has only recently been released and has so far primarily been applied to (small) land-terminating glaciers. Only recently (after we did our experiments with IGM) progress has been made by IGM developers toward modeling tidewater glaciers, e.g. by adding a term that describes the horizontal stress at a calving front (see e.g. the preprint by Jouvet et al. 2024, doi:10.31223/X5T99C). In literature, there is so far only one example of IGM being tested on an synthetic ice shelf (Jouvet and Cordonnier, 2023; doi:10.1017/jog.2023.73), using the SSA approximation (i.e. not higher-order), and no published results exist yet for grounded tidewater glaciers. Main developer of IGM, Guillaume Jouvet, recommended us during a meeting in June this year to perform tests with IGM on tidewater glaciers but was unsure about its performance and potential bugs as no such tests have been performed yet. As we are also curious about how well IGM would do we have now performed a test with IGM for all large glaciers around Kongsfjorden (and beyond) in northwestern Svalbard. To allow for direct comparison with the PISM results, we have, as we did with PISM, set a velocity threshold (25 m a$^{-1}$) to distinguish areas where the sliding coefficient is locally optimized (>25 m a$^{-1}$) or where instead a constant viscosity is used (<25 m a$^{-1}$). Furthermore, exactly the same gridded input datasets (velocity, apparent mass balance, surface height, glacier outlines and initial thickness) are used, and simulations are done at the same model

resolution (500 m). Finally, as for the PISM experiment, we calibrate optimal parameters for Θ and a mass balance correction factor. The figure below shows a comparison of the thicknesses estimated with PISM and IGM. When comparing both maps with available thickness data, we find a slightly better performance with PISM (RMSE = 86 m) than with IGM (RMSE = 92 m) for these glaciers.

[Figure]

Based on the above we have decided that currently it is not worth it yet to model all glaciers with IGM, although this may change in the near future with ongoing progress with IGM. We hypothesize that the reason for the slightly worse performance of IGM on large glaciers (despite the better underlying physics) is that IGM is a machine learning model which does not produce the exact output that a conventional higher order model would produce. It is notably faster, and may through internal learning come close to the results of a higher-order model, but it is not an exact copy. Jouvet and Cordonnier (2023) further noted that IGM experiences a loss of accuracy with increasing domain size, which is another confirmation that IGMs output is not a replica of regular higher-order model results.Other differences between PISM and IGM are in the numerical implementation of e.g. mass fluxes and boundary conditions, which could potentially explain some of the differences even though we are not well enough introduced into both models to give definite answers.

This is more a response to a comment by the other reviewer, but we share it here too as it is related. In general, we are not fully convinced that we should strive for more consistency between the methods for small and large glaciers. We in fact want the methods to be separately optimized for both glacier classes, as it ultimately lowers the thickness errors. What also plays a role here is that there is a physical limit to the degree of detail in the bed that still gives a surface expression (e.g. Gudmundsson et al. 2008). This typically implies that bed features smaller than the ice thickness can not be recovered through inversion. As a result of this we can expect to recover more detailed beds for small glaciers than for large glaciers, and want to modify inversion parameters accordingly.

We have added the following in Sect. 4.2: "*This confirms that the use of IGM for small glaciers leads to better agreement with thickness measurements. One reason may be the higher-order physics behind IGM, which helps to resolve small-scale ice flow and bed*

*features better than with a model like PISM which is based on shallowness assumptions. IGM is under constant development, and to date no extensive tests have been performed yet on grounded tide-water glaciers. Using IGM and the same input datasets and model assumptions as with PISM we performed first tests on a selection of tidewater glaciers in Svalbard showing slightly worse performance (more details in Response to Reviewer 1). This may lie in the machine-learning character of IGM, which can only approximate the results of conventional ice flow models that directly solve the stress equations. It is also worth noting that IGM experiences a loss of accuracy with increasing domain size (Jouvet and Cordonnier, 2023), further underscoring that IGMs output is not a replica of regular higher-order model results.*"

The following was added in Sect. 4.3: "*Arguably, using different ice flow models, spatial resolution, and individual parameter calibration per glacier class some consistency between the methods is lost. However, advantageously we achieve a lower misfit with thickness observations. We further note that there is a limit to the degree of detail in the bed that can be recovered from inversion, which scales with the ice thickness (Gudmundsson et al. 2008). Hence, smaller-scale bed details can theoretically be recovered for smaller (thinner) glaciers than for larger (thicker) glaciers. This supports our use of different resolutions and inverse method calibration for different glacier sizes.*"

Last but not least, you need to say something about the IGM capabilities of inferring ice thickness itself without integration into a transient assimilation. Please add this to the discussion.

We are not sure what the reviewer refers to here. After all, what we are doing is using IGM to infer ice thicknesses without directly assimilating thickness observations, and we show both validation statistics and plots in several places in the manuscript. Our best guess is that the reviewer asks for mentioning that IGM comes with a built-in inversion scheme (which we do not use), and that this should be mentioned in the manuscript. We have therefore added in Sect. 3.2: "*The method is closely aligned with Frank and van Pelt (2024). Note, therefore, that while we use IGM as a forward model, we do not use IGM´s built-in inversion as described by Jouvet (2022) which, in contrast to our method, assimilates thickness observations and relies on cost function minimization.*"

PROGRESS & DISCUSSION
As much as I like it that a new thickness map of the Svalbard ice cover is presented, I wonder about the improvements with respect to existing products. You can either show that the quality of the input data is higher or that your method is more sound. Alternatively, you simply show some performance measures by comparing the different thickness products and leave the decision to the reader. I am not sure what is the best way forward in your case. The most practical is to extend your discussion by additional analysis. I suggest that you simply add previous results to some existing figures (Figs. 5, 6 and 7). This extension will help readers to better assess your new thickness product. In case you find further arguments to promote your map product, please stress this prominently throughout the text and certainly in the abstract. One last thought that got stuck in my head: the two previous estimates (Fürst & Millan) seem to differ in volume and spatial distribution. Your map reproduces the volume of one approach and the pattern of the other. Why is that? Could this give an argument?

Thanks for this comment. We agree the strengths of our approach could be emphasized more. In the original manuscript, we did already quantify how well our study as well as Millan et al. (2022) performed against thickness observations. This information was (is) in Table 2 and showed that our study yields comparable results for small glaciers and a marked improvement for large glaciers and surging glaciers compared to Millan et al. Such a comparison is unfortunately not possible with the results of Fürst et al. (2018) who use an approach that more or less imprints the local thickness observations in their final map.

We do not think it is worth including a comparison (with Millan et al.) in all figures, but do add the scatter plots of modeled vs observed ice thickness also for Millan et al. to Figure 7 (panels c and d). These figures show that the larger MAE for glacier classes 2) and 3) in Millan et al. are a result of a general larger spread and more outliers, especially for larger thicknesses. For glaciers of class 1), i.e. the small ones, Millan et al. tends to underestimate large thickness and overestimate small thicknesses, which is a sign that their thickness distributions are too smooth. We have added sentences on this to Sect. 4.2.

Furthermore, based a related minor comment below, we have now updated the map of Fürst et al. to a newer version (v1.1) that was uploaded to the Norwegian Polar Institute data portal some months after the 2018 study was published (https://doi.org/10.21334/npolar.2018.57fd0db4**)**. As described on the website, the new map applies "averaging in terms of ice viscosity" and has a ~20% higher volume than the original product. The updated Fürst et al. thickness and difference map in Fig. 8c and 8d show that now the volume and average thickness are larger than in our product (7213 vs 6800 km3), and that local differences are much smaller than with the original Fürst et al. (2018) product (v1.0), which showed the much smaller thicknesses compared to our study primarily for the smaller glaciers and low-elevation valleys. We have revised Sect. 4.2 to briefly describe these new results. An independent thickness observation dataset would be needed to compare performance of our study with the product by Fürst et al. which to date is unfortunately not possible. It is noteworthy though that the Fürst et al. products could be seen as an "interpolation method" as the observations are imprinted in the map and mass conservation and viscosity tuning are applied to generate thicknesses in between observations. Our study is less informed by the observations (only to constrain global parameters) which we argue leads to a map that may be more consistent in space (in terms of spatial detail/roughness and uncertainty) and has the advantage that it can be used as a numerically stable spin up state for prognostic modeling. We have added some discussion on this to Sect. 4.2.

INPUT DATA
I would appreciate a little regional overview of all input data in Sect. 2. Many of these data sets have global coverage and it is difficult to assess their quality on Svalbard. You should comment on that briefly. Admittedly, you have Fig. 1 but nothing is said about uncertainties/quality. Possibly some of them are better suite than what was used for previous Svalbard maps.

Admittedly, the input data description was rather short and lacked a discussion of how data were selected. We have added the following to Section 2 to clarify the selection process:

*"The main criteria for the selection of input datasets were: 1) performance in previous comparisons (when available), 2) the time-stamp, since data from a similar period were preferred (see also Sect. 4.3) , and 3) smoothness / spatial noise and missing data. To support the selection of velocity and surface height change datasets we additionally performed tests forcing the inverse method with different products (Millan et al. 2022, Friedl et al. 2021, and NASA ITS_LIVE for velocity; and Morris et al. 2020 and Hugonnet et al. 2021 for surface height change) revealing best performance against thickness data when using Millan et al. (2022) and Hugonnet et al. (2021) respectively. For glacier outlines, we used version 6.0 of the RGI outlines (instead of the newer version 7.0) based on the compatibility of the outline dataset with frontal ablation estimates in Kochtitzky et al. (2022). Differences between the RGI versions 6.0 and 7.0 are in the delineation of individual glaciers, the combined area and the total outline are the same in both versions (see http://www.glims.org/rgi_user_guide/regions/rgi07.html)."*

CALIBRATION
If I understand your manuscript correctly, you calibrate several parameters (spatially uniform) for both the IGM and the PISM inversion. IGM and PISM only differ in the a-priori choice of the rate factor and the sliding coefficient as well as in the SMB correction strategy. Actual calibration parameters are β, Θ, τ and αmin. These mostly relate to the initial guess for ice thickness and the iterative inversion procedure. I cannot understand why you would need different initial guesses for these two models with τPIS M = 0.52kPa vs. τIGM = 100kPa and αPIS M = 0.014 vs. αIGM =0.04. Isn't the perfect plasticity approach only calibrated once?

We understand this is a bit confusing. In general, we argue that because of the different geometries and dynamics of small and large (tidewater) glaciers, there is no reason to assume that the same parameter values would apply.

The reason to use different parameters values for constructing the initial bed is that for IGM the performance (i.e. the agreement with thickness data) was found to be better when using higher values for the two parameters after doing a set of perturbation experiments. For the experiments with PISM it was important to have a smooth transition of the initial and final bed at ice divides between non-surging and surging glaciers (e.g. in ice caps). This was found to be best possible when using the τ and α values for surging glaciers to generate the initial bed for the non-surging glaciers. We have added the following to Sect. 3.2 to clarify that the selection of parameter values for IGM-modeled glaciers (class 1) was based on sensitivity tests:
*"The initial thickness field is obtained using a perfect plasticity approach (eq. (2)) with τ = 100 kPA and αmin= 0.04. These perfect plasticity parameter values were selected based on sensitivity tests with IGM, and hence deviate from the ones used to generate the initial bed for glaciers in class 2 and the final bed for glaciers in class 3."*

Second, I am puzzled why the iterative inversion method requires so different values: βPIS M = 0.25 vs. βIGM = 1 and ΘPIS M = 0.4 vs. ΘIGM = 0.15 (also no friction update in IGM). Can you please explain? These calibration differences cast doubts on keeping a consistent map product while applying two glacier-system models.

β needs to be chosen small enough to avoid too fast corrections of the bed, but not too small to avoid excessive number of iterations. In the end, a too small beta does not impact the final result much (if at all), it will just take more time (iterations) to get there. So we argue that different β values do not affect consistency between the thickness products. We are unsure why a different beta works best with the two models, we hypothesize it is related to the different spatial resolution and time-stepping in the two models.

Θ controls the magnitude of surface updates, which are needed for 'stability' of the inversion (mostly to avoid endless bed adjustments that do not lead to any resulting surface expression). A higher value of Θ generally leads to a bit smoother bed. In both approaches Θ was varied and optimum values were determined for which the thickness error was smallest. This information was missing for IGM and has now been added in the revision. Also, we should note that we mistakenly wrote $\Theta_{IGM} = 0.15$ whereas its actual value was 0.25. This has now been corrected.

We have added the following to Sect. 3.2: *"Whereas β affects the magnitude of bed corrections and number of iterations needed, it hardly (if at all) influences the final bed; a too high value may however cause instabilities and values in PISM and IGM have been chosen accordingly. As in PISM, the value for Θ in IGM has been optimized by minimizing discrepancies with thickness observations."*

MINOR COMMENTS
L91 Please clarify why you chose the perfect plasticity approach for surge-type glaciers. In my view, your main reason is the temporal consistency of the input data. The method by itself is not more adequate for such glaciers. I am not sure if all readers immediately get this.

Indeed, the (lacking) temporal consistency of the input data is the main reason for using the perfect plasticity method. We rephrase as follows:
*"Surging glaciers were modeled separately with a perfect-plasticity method instead, as time-stamp mismatches of the input datasets (e.g. DEM from ~2010 and velocity map from 2017-2018) did not allow for accurate inversion using the Frank et al. (2023) method for glaciers with strong short-term changes in geometry and flow dynamics."*

L107-108 It is not very clear why you limit your surge-type class to the years 2017-2018. Koch et al. (2023) present surging glaciers for a longer time period. Why did you refine your selection? Your DEM dates back to 2010. So many other surges are imprinted. Do you see a problem from that even using the perfect plasticity.

Good point. The difference in timestamps of the input datasets made it difficult to decide which date of period to use to select actively surging glaciers. Our judgment was that the velocity dataset is affected most abruptly by active surging and would cause most uncertainty in the (iterative) inversion, hence we decided to select the collection years for the velocity dataset (2017-2018) as the period to select active surges. With this strategy we hope to have selected the most problematic glaciers, but arguably also glaciers that surged between 2010-2016 are prone to additional uncertainty when modeled with iterative inverse methods. It is however not clear whether the perfect plasticity method would have improved the results for those glaciers. Unfortunately, to our knowledge no complete records exist for

surges between 2010-2016, so this is hard to test. The selection of surging glaciers based on the timestamp of the velocity dataset is described in Sect. 3.3.

L126 From my understanding, the term yield stress relates to when deformation becomes possible whereas the term sliding law normally relates basal velocities to general basal stress conditions. Please check this terminology and be specific about any assumptions. L136 In the in-line equation here, you relate two stresses to each other, namely τd and τc. Velocity is the scaling factor. In this definition, τd and τc cannot both be stresses. I would consider one of them a basal friction coefficient. I therefore would not use the symbol τ.

Thanks for this remark. We use the same terminology as in Bueler and Brown (2009) and in the PISM manual (https://www.pism.io/docs/manual/modeling-choices/subglacier/basal-strength.html).
There was a mistake in the in-line equation, a term describing a threshold velocity was missing in the equation ($u_{thres}$ = 1 m s$^{-1}$). This is now corrected, and with that it is also more clear that the yield stress is indeed a stress and not a coefficient.

L166 Here, you present the first calibrated parameters. Θ represents the strength of the surface update. I am very surprised by its magnitude of 40%. This means that if you have to adjust the thickness at one location, 40% of this change will be imprinted in the surface. This is a lot. I wonder how much your modelled surface then deviates from the observed one after convergence. Please clarify.

Indeed, large surface corrections are applied in locations where the bed deviates much from the initial bed. We forgot to describe in the original manuscript that in PISM this is remedied by first calculating a map of cumulative surface corrections; this map is then smoothed and subtracted from the modeled surface. In PISM this is done once per experiment (halfway, i.e. after 400 iterations), and assures that only minor biases remain in the surface height by the end of the run. This information is now added in Sect. 3.1:
"*To avoid major surface height anomalies relative to the DEM, e.g. when starting from a strongly biased initial bed, we apply a one-time correction to the surface height map after 400 iterations. During this correction, a map of surface height deviations relative to the DEM is computed and smoothed with a Gaussian filter (using four standard deviations for the Gaussian kernel); the resulting map is subtracted from the surface height map.*"

In IGM, no such additional adjustments to the surface height are applied. There is less need because of the lower Θ value that was used, and because a best initial bed was selected by testing initial beds with different parameter settings for the perfect plasticity assumption (see also earlier comment).

L205 & L226 I am happy to see such small threshold values here. Please give them in degrees.

We do not see the benefit of using degrees here. We argue that it only adds to confusion since the 0.001 steps would no longer be equally large.

L219 time periods --> time stamps

Corrected.

L254-256 Your uncertainty estimate of the mean thickness of 205 m is ±7 m . As you say, this is about ±3.5%. Measurement errors of thickness observations typically exceed 10-20% and therefore strongly challenge your estimate. These measurement errors are not considered in your uncertainty analysis. Moreover, I do not get my head around your argument (L255) that the standard deviation of modelled vs. observed thickness values must be divided again by the number of glaciers with observations (i.e., √169) Please explain better or remove this division.

We would like to highlight that the uncertainty estimate (3.5 m) applies to the *mean ice thickness* for all of Svalbard (205 m). It is the mean thickness and its error that are relevant for the ice volume calculation. The fact that the mean thickness bias (i.e. volume error) is zero for 169 glaciers across Svalbard greatly reduces the volume uncertainty for all ice in Svalbard. The volume uncertainty would have been markedly higher when fewer glaciers were observed. Furthermore, the thickness error at a random location in Svalbard is much larger, e.g. in Table 2 it can be found that it is 75.5 m for glaciers in class 2) and 3) and 50.1 m for glaciers in class 1. In other words, local errors that occur at individual sites in Svalbard to a large extent balance / average out at the Svalbard-wide scale. The following has been added to Sect. 3.4:
"*Please note that the relative error of the volume and mean thickness is much smaller than the local (point) uncertainty of modeled thicknesses (the latter is quantified in Sect. 4.2).*"

and in Sect. 4.3:

"*The large and well-distributed thickness observations dataset available for Svalbard used for model calibration, including data from 169 glaciers, helped to reduce the Svalbard-wide volume uncertainty (estimated at 3.5 %). Whereas the RMSE of Svalbard mean glacier thickness is only 3.5 m as a result of averaging and calibration, the local (point) thickness error is considerably larger (50.1 m for class 1 and 75.5 m for class 2 and 3, Tab. 2).*"

The reasoning behind the division with sqrt(169)=13 is that if we would have calibrated the model (i.e. removed the bias) using data from only one arbitrary glacier out of the 169 observed glaciers, we would introduce a bias in the mean thickness after calibration that is 68% of the time between -45 and + 45 m (depending on which glacier we chose to calibrate the model parameters against). This range of biases narrows if we select more than one glacier for calibrating the model, and, following the same logic as is used to calculate a standard error of a mean, it can be found that dividing by the square-root of the number of samples gives the correct the mean thickness error when selecting all 169 glaciers. E.g. in case numerous random selections of 5 observed glaciers were used to calibrate the model (i.e. to remove the thickness bias for those 5 glaciers), the standard deviation of the mean thickness would reduce to +/-45 divided by sqrt(5). We hope this clarifies it a bit and understand it can be rather confusing. In the manuscript we have revised the description of this in Sect. 3.4 by adding:
"*The range of biases narrows if we select more than one glacier for calibrating the model, and, following the same logic as is used to calculate a standard error of a mean, it can be found that dividing by the square-root of the number of samples is required to calculate the remaining standard deviation for larger sets of glaciers used for calibration.*"

L270 For the land- and marine terminating glaciers, you report mean thickness values of 42 m and 162 m, respectively. How can you reconcile this value with the archipelago-wide average of 205 m. I certainly miss something here ;o)

The mean thickness values are correct. The values for land and marine terminating glaciers are *median* values whereas for all glaciers we report the *mean* thickness. Given the skewed distribution with relatively many (very) small glaciers with small thickness, the median values are much lower than the mean.

FIGURES
Fig. 1 I like this figure very much. In panel (c) you indicate the locations of all surge-type glaciers. They all belong to larger ice-fields. So are they now modelled with PISM or the perfect plasticity approach. In the latter case, how do you ensure that you do not get internal fringe lines in the thickness field.

See earlier response.

Fig. 2 Could you add the reference run with $M_{corr}$ = 0.4 m w.e. yr−1 and $\Theta$ = 0.4.

Please note that $M_{corr}$ = 0.4 and $\Theta$ = 0.4 were the reference (best) values that are used for producing the final thickness and bed estimates with PISM (as shown in Fig. 3). Figure 2 shows differences relative to this (best) reference run. We have changed the caption of Fig. 2 to make this more clear and changed a sentence in Sect. 3.1.

Fig. 4 In panel (a) and (b), the ice thickness decreases towards the calving fronts of marine-terminating glaciers. Why is that? On Hansbreen, you could directly compare to a dense survey grid included in GlaThiDa.

Most tidewater glaciers thin towards their fronts. This is because of mass continuity and increased flow rates closer to the calving front, which lead to less thick ice. This is for example obvious on Kronebreen (e.g. Lindbäck et al. 2018), and Hansbreen too (e.g. Oerlemans et al. 2011). Indeed, we could have included a data comparison in Figure 4, (e.g. using colored dots at observation sites), but this would make the figure much harder to read. Since the purpose of the figure is mainly to show local examples of how the 100-m resolution IGM results and 500-m PISM & perfect-plasticity results are merged, we prefer to stick with the current format.

Furthermore, since the Fürst dataset (locally) assimilates all thickness observations, the spatial comparison in Figure 8b and 8d could be used to e.g. look at performance of our thickness product on densely observed glaciers like Kronebreen and Hansbreen.

Fig. 4 & 5 Please extend these figures by values from the existing thickness maps. It might also help in evaluating & promoting your results.

We think that with the additions to Fig. 7, the table with statistics (Table 2), and the (updated) spatial comparison maps in Fig. 8 there is sufficient comparison with previous studies.

Fig. 7 How does this distribution look like for Millan? It would be good to add for reference. It might also help in evaluating & promoting your results.

We are thankful for this suggestion and include two additional panels in Fig. 7 to also validate Millan's product against thickness observations.

We have further added the following in Sect. 4.2: "*Similar scatter plots comparing thicknesses by Millan et al. (2022) with observations (Fig. 7c-d) show that the larger errors for glaciers in classes 2 and 3 (Table 2) are a result of a general larger spread in the Millan et al. (2022) dataset, primarily for large thicknesses. For the small glaciers (class 1) Millan et al. (2022) show an underestimation of large thicknesses and an overestimation of small thicknesses, indicating that the Millan et al. (2022) thickness product is smoother than reality.*"

Fig. 8 Please use the updated thickness map (v1.1) from Fürst et al. (2018) at data.npolar.no.

Thanks for this suggestion. We now include v1.1 and have updated the related discussion in Sect. 4.2. We now more extensively discuss the different nature of the Fürst et al. map, which imprints the observations, and added discussion on advantages and drawbacks of both methods. We also mention the large volume difference (~20%) of version 1.0 and 1.1. The volume estimate for version 1.1 has been added to the Introduction.

TABLES
Table 1 Please indicate the version of GlaThiDa.

We used version 3.1.0, and now include this information in Table 1.

Table 2 Please use the numbering you introduced for the glacier classes (1-3).

Good point, we have revised this here and throughout the manuscript.

CODE AVAILABILITY
Is the inversion method available via an open repository?

The current scripts we used for the inversion are currently not easy to use by others as they lack commenting and the (Python) code is not optimally structured yet. We hence prefer to wait with publishing them until they are more readable. Nevertheless, we believe the Methods section and references therein are sufficiently detailed to allow reproduction of our results. Both ice flow models (IGM and PISM) are open-access, as well as all input datasets. Furthermore, the generated thickness and bed maps are available in a repository (see Data Availability).

---

## Author Comment (AC2)

**Response to RC2:**

Referee comments in **red**
Author comments in **green**

**Summary**

This paper presents a new ice thickness and bed map for Svalbard based on a method previously developed by the authors. Here, they use three different methods for small, land-terminating glaciers (IGM), larger and tidewater glaciers (PISM), and surging glaciers (perfect plasticity) to derive their results, having performed a considerable amount of calibration and processing to reduce errors and ensure agreement across their results (no big jumps in ice thickness at ice divides). They then compare their results to other recent bed-thickness datasets for Svalbard, showing that their work sits within the expected range, but substantially reduces errors and bias across the board.

I think this is an innovative paper that attempts to leverage recent developments to obtain the best-possible results. However, I have a few major concerns, as well as several minor ones before I would be happy to recommend the paper for publication. I wonder whether the use of three different methods, particularly when it is not clear to me the rationale for using two different ice-flow models, has not overcomplicated the paper and sacrificed internal consistency – normally an advantage of these large-scale datasets – for an unclear gain in accuracy. The authors establish using PISM where they use IGM would lead to a worse outcome, but not whether the reverse is true, which seems to me a major oversight that makes it difficult to see what advantage using PISM and complexifying the method really brings. I am also very unclear as to how the authors set up their model domains and how they dealt with the resulting boundary conditions, which makes it difficult for me to assess the quality of their modelling. Overall, I therefore think major revisions are required: it may just be a question of adding/clarifying some information that is not obvious as the paper is currently written, which I hope is the case, as I think the end outcome and method are very interesting!

I should also note that I read the paper and wrote the review before I read Reviewer 1's comments. The fact that both of us largely raise the same issues is therefore not a result of groupthink.

Line and page numbers refer to those in the manuscript.

We are very grateful for the detailed and constructive comments, which have helped us greatly to improve the manuscript!

**Major Points**

- Consistency: this is perhaps something of a more philosophical point, but a major advantage of these kinds of large-scale studies is that (usually) they apply the same processing steps to a wide area so that, even if one doesn't believe the absolute numbers very much, one can be confident that the results are internally self-consistent. Here, the use of three (very) different methods means this cannot be taken for granted in the same way. I think the authors have done a lot of work

to try to overcome this, particularly with interpolating results onto different grids, but I wonder if this was the right approach.

- Choice of methods: I do not think the authors really provide a clear justification for why they chose to use PISM for the larger (and tidewater) glaciers, as opposed to IGM. I take the point that PISM in SIA+SSA mode does a good job for larger and tidewater glaciers, but IGM is a higher-order model that would also do a good job here. Usually, this choice would be justified by saying that higher order models are too computationally expensive to make them practical at this scale, but IGM has been explicitly designed to run fast and overcome that objection. So what motivated the choice? Because, if the authors had used IGM for all the larger and tidewater glaciers, that would have allowed them to achieve a much larger degree of consistency in the results and avoided them a considerable degree of work at the calibration, method and post-processing stage, it seems to me (of course, they would have had to find a strategy to vary the sliding parameter in IGM, but that feels to me an easier thing to do than have two separate methods working on different assumptions at different resolution)

These are very good points that we have given a lot of thought before submission, but also in recent weeks. Since similar questions were raised by the other reviewer, we post a copy of that response below.

Ideally all glaciers should be modeled with the best possible physics. Following that logic it would have made sense to model also the large tidewater glaciers using IGM. There are however several reasons for us to choose PISM instead for the large glaciers. PISM is an ice flow model that has previously been successfully applied to model glaciers and ice caps that exhibit both sliding and non-sliding flow (by combining the shallow shelf and shallow ice approximations). IGM has only recently been released and has so far primarily been applied to (small) land-terminating glaciers. Only recently (after we did our experiments with IGM) progress has been made by IGM developers toward modeling tidewater glaciers, e.g. by adding a term that describes the horizontal stress at a calving front (see e.g. the preprint by Jouvet et al. 2024, doi:10.31223/X5T99C). In literature, there is so far only one example of IGM being tested on an synthetic ice shelf (Jouvet and Cordonnier, 2023; doi:10.1017/jog.2023.73), using the SSA approximation (i.e. not higher-order), and no published results exist yet for grounded tidewater glaciers. Main developer of IGM, Guillaume Jouvet, recommended us during a meeting in June this year to perform tests with IGM on tidewater glaciers but was unsure about its performance and potential bugs as no such tests have been performed yet. As we are also curious about how well IGM would do we have now performed a test with IGM for all large glaciers around Kongsfjorden (and beyond) in northwestern Svalbard. To allow for direct comparison with the PISM results, we have, as we did with PISM, set a velocity threshold (25 m a$^{-1}$) to distinguish areas where the sliding coefficient is locally optimized (>25 m a$^{-1}$) or where instead a constant viscosity is used (<25 m a$^{-1}$). Furthermore, exactly the same gridded input datasets (velocity, apparent mass balance, surface height and glacier outlines) are used, and simulations are done at the same model resolution (500 m). The figure below shows a comparison of the thicknesses estimated with PISM and IGM. When comparing both maps with available thickness data, we find a slightly better performance with PISM (RMSE = 86 m) than with IGM (RMSE = 92 m) for these glaciers.

[Figure]

Based on the above we have decided that currently it is not worth it yet to model all glaciers with IGM, although this may change in the near future with ongoing progress with IGM. We hypothesize that the reason for the slightly worse performance of IGM on large glaciers (despite the better underlying physics) is that IGM is a machine learning model which does not produce the exact output that a conventional higher order model would produce. It is notably faster, and may through internal learning come close to the results of a higher-order model, but it is not an exact copy. Jouvet and Cordonnier (2023) further noted that IGM experiences a loss of accuracy with increasing domain size, which is another confirmation that IGMs output is not a replica of regular higher-order model results. Other differences between PISM and IGM are in the numerical implementation of e.g. mass fluxes and boundary conditions, which could potentially explain some of the differences even though we are not well enough introduced into both models to give definite answers.

In general, we are not fully convinced that we should strive for more consistency between the methods for small and large glaciers. We in fact want the methods to be separately optimized for both glacier classes, and it ultimately lowers the thickness errors. What also plays a role here is that there is a physical limit to the degree of detail in the bed that still gives a surface expression (e.g. Gudmundsson et al. 2008). This typically implies that bed features smaller than the ice thickness can not be recovered through inversion. As a result of this we can expect to recover more detailed beds for small glaciers than for large glaciers, and want to modify inversion parameters accordingly.

We have added the following in Sect. 4.2: "*This confirms that the use of IGM for small glaciers leads to better agreement with thickness measurements. One reason may be the higher-order physics behind IGM, which helps to resolve small-scale ice flow and bed features better than with a model like PISM which is based on shallowness assumptions. IGM is under constant development, and to date no extensive tests have been performed yet on grounded tide-water glaciers. Using IGM and the same input datasets and model assumptions as with PISM we performed first tests on a selection of tidewater glaciers in Svalbard showing slightly worse performance (more details in Response to Reviewer 1). This may lie in the machine-learning character of IGM, which can only approximate the results of conventional ice flow models that directly solve the stress equations. It is also*

*worth noting that IGM experiences a loss of accuracy with increasing domain size (Jouvet and Cordonnier, 2023), further underscoring that IGMs output is not a replica of regular higher-order model results."*

The following was added in Sect. 4.3: "*Arguably, using different ice flow models, spatial resolution, and individual parameter calibration per glacier class some consistency between the methods is lost. However, advantageously we achieve a lower misfit with thickness observations. We further note that there is a limit to the degree of detail in the bed that can be recovered from inversion, which scales with the ice thickness (Gudmundsson et al. 2008). Hence, smaller-scale bed details can theoretically be recovered for smaller (thinner) glaciers than for larger (thicker) glaciers. This supports our use of different resolutions and inverse method calibration for different glacier sizes.*"

- Boundary conditions and model domains: I am unclear how the authors defined their model domains. I assume, given they use RGI6.0, that they take each RGI outline and invert it individually? Or do they take all contiguous RGI6.0 outlines and invert them as one entity? In the former case, how do they then deal with ice-ice boundaries, where two different RGI entities are in contact? In both cases, what boundary condition is imposed at the front of tidewater glaciers? As both the PISM and IGM methods involve small forward timesteps, these issues need to be considered. At the very least, a few lines in the discussion about how not considering these likely introduces some local inaccuracies need to be added.

Thanks for this comment, we realize it was not clearly explained in the original manuscript. In PISM, all large glaciers (class 2), actively surging glaciers (class 3) and small glaciers (class 1) that are part of large connected ice systems were modeled in one go. The thickness of surging glaciers, from the perfect plasticity assumption, was held fixed during the simulation, but mass exchange at the ice divides was possible. In IGM, glaciers are generally modeled individually, but glaciers that are connected to other glaciers are modeled in one go. The above approach avoids thickness jumps both between glaciers in classes 2 and 3 in PISM, and between connected glaciers in class 1 modeled with IGM. Finally, to avoid that large ice systems consist of glaciers of class 1 (modeled with IGM) and 2 (modeled with PISM), which would create jumps between them, we decided to not use IGM but rather PISM output for small glaciers within those large ice systems.

At the front of tidewater glaciers, we simply assume all ice to calve off that flows out of the outline. So glaciers cannot advance beyond the outline. Given the positive apparent mass balance of tidewater glaciers and the mass balance correction term (to compensate for mass lost through side boundaries), fronts of tidewater glaciers generally have no tendency to retreat either.

The following is reformulated / added to Sect. 3 (first paragraph):
"*One nuance to the three groups above is that all (small) glaciers in class 1 that are part of / connected to larger ice caps are modeled with PISM. This is to avoid thickness jumps at the ice divides. Furthermore, to avoid thickness jumps within ice caps between PISM-modeled and surging glaciers, experiments with PISM also include the surging glaciers as static entities with thicknesses based on the perfect-plasticity assumption.*"

and this is added in Sect 3.1:

"*The positive apparent mass balance for tidewater glaciers together with a positive $M_{corr}$ commonly assure a positive mass flux (i.e. calving / frontal ablation) at the calving front. Hence, calving fronts do not retreat. They do not advance either since all mass that flows out of the outlines defined by the RGI dataset is instantly removed.*"

> • Discussion: Ultimately, I think this comes back to my point on the choice of methods above, but I don't find that the discussion does a very good job of highlighting what this study brings to the table and why people should use the bed calculated in this study as opposed to those from other studies. The authors provide plenty of description for how their results compare to other datasets, but mostly do not analyse why these differences occur, making it hard for readers to assess which product is better for their particular application. The fact that it is also unclear as to why the authors made particular methodological choices (see my comment above) then further muddies the waters here. The authors do show that they substantially reduce the error on larger glaciers compared to previous studies and the bias across all glaciers (Table 2), which I would argue is the main selling point of their results in the current formulation of the paper, but this gets a bit lost in the discussion and no mention of it is made in the abstract (there is a partial reference in the conclusion, but only to the error on larger glaciers), making it very easy for readers to lose sight of it completely.

Thanks for this comment. We agree the strengths of our approach could be emphasized more.

First of all, we see the use of dedicated methods for small, large and surging glaciers as a strength of our work, and we hope the changes made in response to the first major comments on consistency and choice of methods has helped to better emphasize this.

Furthermore, in the original manuscript, we did already quantify how well our study as well as Millan et al. (2022) performed against thickness observations. This information was (is) in Table 2 and showed that our study yields comparable results for small glaciers and a marked improvement for large glaciers and surging glaciers compared to Millan et al. Such a comparison is unfortunately not possible with the results of Fürst et al. (2018) who use an approach that more or less imprints the local thickness observations in their final map. An independent thickness observation dataset would be needed to compare performance of our study with the product by Fürst et al. which to date is unfortunately not possible. It is noteworthy though that the Fürst et al. approach can be seen as an "interpolation method" as the observations are imprinted in the map and mass conservation and viscosity tuning are applied to generated thickness in between observations. Our study is less informed by the observations (only to constrain global parameters) which we argue leads to a map that may be more consistent in space (in terms of spatial detail/roughness and uncertainty) and has the advantage that it can be used as a numerically stable spin up state for prognostic modeling. We have added discussion on this to Sect. 4.2:
"*It is noteworthy though that the Fürst et al. products can be seen as an 'interpolation method' as the observations are imprinted in the map and mass conservation and viscosity tuning are applied to generated thickness in between observations. Our study is less*

*informed by the observations (only to constrain global parameters) which we argue leads to a map that may be more consistent in space (in terms of spatial detail/roughness and uncertainty) and has the advantage that it can be used as a numerically stable spin up state for prognostic modeling.*"

Additionally, after related comments by the other reviewer, we have added scatter plots of modeled vs observed ice thickness also for Millan et al. to Figure 7 (panels c and d). These figures show that the larger MAE for glacier classes 2 and 3 in Millan et al. are a result of a general larger spread and more outliers, especially for larger thicknesses. For glaciers of class 1, i.e. the small ones, Millan et al. tends to underestimate large thickness and overestimate small thicknesses, which is a sign that their thickness distributions are too smooth. We have added sentences on this to Sect. 4.2:

"*Similar scatter plots comparing thicknesses by Millan et al. (2022) with observations (Fig. 7c-d) show that the larger errors for glaciers in classes 2 and 3 (Table 2) are a result of a general larger spread in the Millan et al. (2022) dataset, primarily for large thicknesses. For the small glaciers (class 1) Millan et al. (2022) show an underestimation of large thicknesses and an overestimation of small thicknesses, indicating that the Millan et al. (2022) thickness product is smoother than reality.*"

**Minor Points**

- p.1, l.10: I might venture to say that a mean ice thickness at the scale of the whole Svalbard archipelago isn't that useful or meaningful a number to include in the abstract (at least, as a headline figure for the paper, it seems some way down the list of things that readers would want to know)? The total volume, yes, but I would suggest maybe converting that into an SLR equivalent for the second number, or reporting the maximum ice thickness, which is something that makes a bit more sense at that scale. Or, possibly even more useful, say something about how the volume estimate presented here compares to other studies' estimates.

Thanks for this comment. We have now removed the mean thickness and instead added the sea level equivalent to the abstract. Furthermore, the sea level equivalent estimate has been added to Sect. 4.1.

- p.2, l.42: Reference formatting for Farinotti et al.

This is now corrected.

- Table 1: Why use the 20 m NPI DEM when it has to be downscaled to 100 or 500 m immediately? Wouldn't the COP90 DEM have been a better choice to fit with the modelling resolutions and also sit more in the middle of the range of most of the other data (2010-2019ish)? Also, the RGI6.0 outlines for Svalbard have dates of 2000-2010, so please update the table to reflect that.

Good point, although most of the outlines are from 2007-2008, there are some outlines from other years (e.g. 2001). We have changed the period in Table 1 to 2000-2010 as suggested. We have chosen the NPI DEM over global DEMs such as the COP90 DEM, since it is a

dedicated product for Svalbard that incorporates a large amount of recent and older data from aerial photography (https://doi.org/10.21334/npolar.2014.dce53a47). Furthermore, the error (standard deviation) of the NPI DEM has been quantified (2-5 m, possibly slightly larger on glaciers), whereas such information for Svalvard is, to our best knowledge, missing for e.g. the COP90 DEM. Finally, in response to the other reviewer we have extended the input data description in Sect. 2.

- Figure 1: I can see already that, on Austfonna, there are two methods being used to generate the results, despite the assertion in lines 116-117 that all the glaciers connected to larger ice caps are modelled using PISM (i.e. the same method). Can the authors confirm whether the figure or the text is correct here? If the figure is right, how are they dealing with the jumps in thickness at the ice divides on Austfonna?

See also our earlier reply. Our statement in the original manuscript that PISM is used for all glaciers that are part of larger ice systems was incorrect. Surging glaciers that are part of large ice systems were modeled with the perfect plasticity assumption. The surging glaciers were included in the PISM model runs (as static entities) to provide boundary conditions at ice divides for the modeling of the non-surging glaciers.

- p.7, l.169: 'with a'

Corrected.

- p.9, l.200-204: Can the authors comment as to how far using uniform parameter values might introduce some error into the results?

Using uniform parameters here is foremost a practical choice as a result of 1) the observational datasets of ice velocity being of too low quality for slow-flowing mountain glaciers to deliver a reliable signal that could be used for a spatially variable sliding coefficient inversion as in our PISM approach; 2) the sample size of glaciers in class 1 with observations not being big enough to deduce any spatial/climatic patterns of A and c that could be extrapolated to unsurveyed glaciers. Not the least, the complex poly-thermal nature of many Svalbardian glaciers is a complicating factor. As such, the calibrated values for A and c are on average the best fitting ones, but naturally there may be glaciers with specific local conditions where they are not ideal. However, we do not see a feasible way of systematically classifying where this would be the case. Consequently we are left to conclude that any errors resulting from the spatially homogenous A and c values likely are included in the overall uncertainty (Table 2), given that the calibration glaciers form a fairly representative sample of all glaciers.

- p.9, l.214: I confess I'm not entirely clear on why the thickness field produced by IGM would have gaps in it that need interpolating?

We correct for the mass leaking out of the glacier domain with a spatially uniform adjustment of the specific mass balance. However, also some part of this mass addition can be leaking out, which then means that (usually small) parts of the domain remain ice free. These holes we interpolate in the end. This reasoning is also described in Frank and van Pelt (2024) which we cite. To clarify, we have rephrased line 214 as follows: "*The final thickness field is*

*obtained by interpolating gaps in the modeled thicknesses which may remain in the case of persistent mass leaking and applying a thickness-dependent Gaussian filter as in Frank and van Pelt (2024)."*

> • p.10, l.231-232: This seems to me quite a substantial upsampling of the majority of the dataset that might introduce a considerable number of artefacts. Would not downsampling the 100 m proportion of the dataset to 500 m have been the more conservative choice?

Thanks for bringing this up. By using nearest neighbor interpolation when reprojecting the 500-m results to a 100-m resolution grid, we do not add any detail to the bed (it will look exactly the same on a 100 and 500 m grid). Furthermore, we prefer to keep the 100-m results at their original resolution so that no detail is lost there. We would also like to repeat that we believe it is justified to have finer resolution bed topography for small (thin) glaciers than for large (thick) glaciers (see our response to the first major comments).

> • p.10, l.245-257: Yes, but are there observed glaciers in each of the three categories, such that all three types of inversion are bias-free? More generally, with three different inversion methods, would there not need to be three separate estimates of sigma H bar, one for each category? Because a mean error of 3.5 m on ice thickness across the whole of Svalbard seems a little too good to be true. The observations themselves would have bigger errors than that!

We would like to highlight that the uncertainty estimate (3.5 m) applies to the *mean ice thickness* for all of Svalbard (205 m). It is the mean thickness and its error that are relevant for the ice volume calculation. The fact that the mean thickness bias (i.e. volume error) is zero for 169 glaciers across Svalbard greatly reduces the volume uncertainty for all ice in Svalbard. The volume uncertainty would have been markedly higher when fewer glaciers were observed. Furthermore, the thickness error at a random location in Svalbard is much larger, e.g. in Table 2 it can be found that it is 75.5 m for glaciers in class 2) and 3) and 50.1 m for glaciers in class 1. In other words, local errors that occur at individual sites in Svalbard to a large extent balance / average out at the Svalbard-wide scale. The following has been added to Sect. 3.4:
*"Please note that the relative error of the volume and mean thickness is much smaller than the local (point) uncertainty of modeled thicknesses (the latter is quantified in Sect. 4.2).*

And in Sect. 4.3:

*"The large and well-distributed thickness observations dataset available for Svalbard used for model calibration, including data from 169 glaciers, helped to reduce the Svalbard-wide volume uncertainty (estimated at 3.5 %). Whereas the RMSE of Svalbard mean glacier thickness is only 3.5 m as a result of averaging and calibration, the local (point) thickness error is considerably larger (50.1 m for class 1 and 75.5 m for class 2 and 3, Tab. 2)."*

Furthermore, we argue that splitting the glaciers into separate categories for the uncertainty assessment would only complicate the uncertainty assessment and is unlikely to lead to a very different error estimate for the mean thickness. The error estimate would only differ significantly if the relative area fraction of observed glaciers in a certain class differ from the relative area fraction of the same class for all glaciers in Svalbard. Since observations are

well-spread over Svalbard and include glaciers of all types, we assume this effect would be small. Please also note that rather than splitting between glacier classes we could also group the glaciers in other categories with individual errors (e.g. creating categories of thin and thick glaciers, or tide-water and non-tidewater glaciers, or to split Svalbard in regions). All would give a slightly different Svalbard-wide error estimate (sometimes higher, sometimes lower) than when lumping all observed glaciers together.

To better clarify the uncertainty assessment strategy we have reformulated some sentences in Sect. 3.4:
"*The range of biases narrows if we select more than one glacier for calibrating the model, and, following the same logic as is used to calculate a standard error of a mean, it can be found that dividing by the square-root of the number of samples is required to calculate the remaining standard deviation for larger sets of glaciers used for calibration.*"

- p.16, l.314-319: Have the authors performed the same comparison in the other direction? As in, what happens if IGM is used to model the larger glaciers where PISM was the preferred method? Otherwise, I think it's difficult to say that the combination of the two methods is superior to either alone. The approximations in PISM may be suitable on the larger glaciers, but it doesn't follow that that means they're more suitable than using a higher-order model.

Thanks for this comment. We refer to our response to the first major comment.

- p.16, l.321: OK, yes, 6855 is higher than 6800 and 207 is higher than 205, but I'm not sure that it's really a meaningful difference, especially when both those numbers are well within this study's own error bars. Consider rephrasing this to make it clearer that this study's integrated volume and mean thickness results are not significantly different to those from Millan et al.

We agree, this is now corrected.

- p.16, l.320-333: Could the authors provide some more analysis of why these differences exist? They posit sensible reasons for why Millan et al. likely overestimate ice thickness on larger glaciers, but I think it would make the paper much more useful for the community if they can suggest some reasons for the other differences (spatial distribution more similar to Fürst, thicker ice at lower elevations than Fürst, less pronounced jumps at ice divides than Millan, etc.), as it would help people work out which is the best bed product for them to use for their particular application

We refer to our response to the last major comment above.

- p.18, l.384: This is only true provided other people use the exact same set of final modelled bed, velocity, surface, etc. as used in this study as their initial conditions. If someone took the bed from this study and then used, say, the COP90 surface DEM and ITSLive velocities to initialise their model, they would not have a harmonious set of initial conditions. Please rephrase this to make it more clear.

Thanks for pointing this out. We have rephrased the sentence to make it clear that it is general benefit of this type of inverse methods (i.e. iterative ones) that the reconstructed beds are a starting point for potential future runs when using the same model, setup and compatible input datasets:

"*A benefit of thickness maps produced with iterative inverse methods, i.e. for all not actively surging glaciers, is that they simultaneously provide initial conditions for future simulation of the same set of glaciers. However, this does require the use of the same ice flow model, setup, and temporal consistency of input datasets.*"

---

## Referee Report (RR1)

**Review of van Pelt and Frank (2024) 'New glacier thickness and bed maps for Svalbard'**

**Summary**
I reviewed the first iteration of this paper and recommended major revisions based on several areas of method and discussion in the original manuscript that required clarification to be understandable. I am very pleased that the authors took these comments on board in a very thorough response and set of revisions to the manuscript that deal well with my earlier concerns. As I'd hoped, it was primarily a case of more explanation required rather than anything else. Consequently, I have no further issues with the paper (beyond a few very small typos) and can recommend it for publication. Congratulations to the authors on their work!

Page and line numbers refer to the clean version of the revised manuscript.

**Major Comments**
None

**Minor Comments**
- p.17, l.350: 'help' not 'helps' if I'm being very picky.
- p.18, l.377: 'the generated thicknesses between the observations'
- p.19, l.378: 'only used to constrain global parameters'

---

## Referee Report (RR2)

Comments on the revised manuscript
egusphere-2024-1525 entitled
**New glacier thickness and bed maps for Svalbard**

presented on 03.09.2024
by

Ward Van Pelt and Thomas Frank

The authors present a revised manuscript addressing the comments
from the initial review round.  Although I appreciate the authors
effort to answer the major comments, I regret to say that some of
their answers to several of my major concerns are not convincing
- at least to me.  I still remain very positive about this manuscript
and therefore recommend that the editor should continue to considered
it for publication in *The Cryosphere* after my concerns have been
alleviated.

**MAJOR COMMENTS**

**METHODOLOGICAL MIX**
Thank you for including the comparison between PISM and IGM including
RMSE values and thickness maps in the rebuttal (not in the manuscript).
You argue that PISM performs better for regional application.  The
fact why you prefer IGM over PISM for small-scale glaciers remains
vague (higher-order) and not convincing to me.  I still do not see
the ultimate argument to included IGM - certainly after you invoke
the limitations of the current fast developments.  I still wonder
why you are not more consistent and apply PISM all over the domain
(with prescribed perfect-plasticity values for surging glaciers).
In this way, you would get a more coherent map product and a simpler
method (also in terms of calibration).

**PROGRESS**
In response to this comment, you raise the argument that your approach
*'can be used as a numerically stable spin up state for prognostic
modeling'*.  I agree that this initialisation is big asset.  Yet
the built-in IGM inversion for thickness and ice-flow parameters
claims the same property.  Please discuss.  Do not misunderstand
me, I can accept this argument as a clear benefit.  Yet, I imagine
myself to start modelling on Svalbard with your thickness map.  I
would then need to use IGM and PISM to do so consistently.  It might

be me, but it seems impractical to do so with two models that need
to communicate/interact. Again, the solution is to reduce the method
mix to PISM-only (with perfect plasticity).

**PERFECT PLASTICITY**
Your answers to my minor comments on L91 and L107-108 are not yet
satisfying. You agreed that temporal consistency motivated your
choice - specifically with regard to the 2010 DEM and 2017-2018
surface velocity observations. Following this logic, you should
rather use the Copernicus DEM. Moreover, the perfect plasticity
does not require surface velocities. So your argument should invoke
the glacier outlines and the DEM. This also corroborates your argument
on limiting the surge-type identification to the velocity fields
from 2017-18. As I indicated before, the analysis of Koch et al.
(2023) shows an extended time coverage for surge-type glacier identification.
Finally, you did not comment on the applicability of the perfect
plasticity approach for surge-type glaciers. Please add.

**MINOR COMMENTS**

**L126** 'This comment was undressed (neither in the rebuttal nor in
the revised manuscript).' From my understanding, the term yield
stress relates to when deformation becomes possible whereas the
term sliding law normally relates basal velocities to general basal
stress conditions. Please check this terminology and be specific
about any assumptions.

**L270** 'Follow-up on your answer to:' For the land- and marine terminating
glaciers, you report mean thickness values of 42 m and 162 m, respectively.
How can you reconcile this value with the archipelago-wide average
of 205 m. I certainly miss something here ;o)

Obviously, I missed the difference between median and mean values.
This metric-mix makes it very hard to compare. Solutions would
be to either use both metrics for each time you specifically provide
averaging measures or only give either of the two metrics consistently
throughout the manuscript. Personally, I prefer the former. Your
decision.

---

## Author Response (AR2)

**Response letter**

*Referee comments in black.*
*Author comments in green.*

**Response to Reviewer #1**

We thank the reviewer for another evaluation of our manuscript and are glad to hear that all comments were satisfactorily addressed.

**Response to Reviewer #2**

The authors present a revised manuscript addressing the comments from the initial review round. Although I appreciate the authors effort to answer the major comments, I regret to say that some of their answers to several of my major concerns are not convincing - at least to me. I still remain very positive about this manuscript and therefore recommend that the editor should continue to considered it for publication in The Cryosphere after my concerns have been alleviated.

We are grateful to the reviewer for taking the time and effort to go through the manuscript again. We are glad the reviewer is still positive about the manuscript and regret that we were not able to take away all the concerns with our previous responses. With the new revisions and replies below we hope to have satisfactorily addressed the remaining concerns!

MAJOR COMMENTS

METHODOLOGICAL MIX
Thank you for including the comparison between PISM and IGM including RMSE values and thickness maps in the rebuttal (not in the manuscript). You argue that PISM performs better for regional application. The fact why you prefer IGM over PISM for small-scale glaciers remains vague (higher-order) and not convincing to me. I still do not see the ultimate argument to included IGM - certainly after you invoke the limitations of the current fast developments. I still wonder why you are not more consistent and apply PISM all over the domain (with prescribed perfect-plasticity values for surging glaciers). In this way, you would get a more coherent map product and a simpler method (also in terms of calibration).

Thank you for this comment. After the first review round we mistakenly assumed the reviewer was only wondering why we did not use IGM everywhere (possibly because of the focus on that in the other review), and hence did not discuss the option whether to use PISM everywhere. In our previous response we addressed why PISM is preferred over IGM for the large (tidewater) glaciers, which is primarily because IGM has not yet been tested extensively on such glaciers. And we found worse performance of IGM for these glaciers in a first test in northwestern Svalbard.. For small land-terminating glaciers we rather find the opposite. IGM has in various applications on (small) mountain glaciers shown strong performance (e.g. Cook et al. 2023; Jouvet and Cordonnier, 2023; Jouvet et al. 2022) and its low computational cost and higher-order physics make it a suitable model for high-resolution modeling of small (steep) glaciers. We argue from an ice flow physics point of view that the

benefit of using higher-order physics for small (often steeper) mountain glaciers is larger than for larger thick (relatively flat) tide-water glaciers and ice caps. This is because the shallowness assumptions (SIA and SSA) become less accurate for the typically larger depth-to-width ratios of mountain glaciers. Besides this, our main argument to use IGM for small glaciers is that it is simply performing better than PISM for these glaciers. We have modeled the small glaciers also with PISM (at 500-m resolution) and a comparison of the related statistics reveals that using PISM instead of IGM leads to an increase of the MAE from 38.0 to 42.7 m and RMSE from 50.1 to 54.1 m. Furthermore, the R-correlation drops from 0.77 (IGM) to 0.71 (PISM). To allow for a direct comparison we have first reprojected the 500-m PISM results to the finer 100-m grid used by IGM using nearest-neighbor interpolation.

We agree with the reviewer that the use of two instead of one ice flow model reduces the coherency of the results of glaciers in classes 1 and 2. However, our aim is to produce a best possible thickness map, which as shown requires the use of different flow models. In a similar fashion we justify the use of two different resolutions for glaciers in class 1 (100-m) and class 2 and 3 (500-m). This also reduces coherency between the results of the two classes but we value the greater detail in the final thickness and bed maps more than the downside of having inconsistent resolutions across the thickness and bed maps.

We reformulated and added the following in Sect. 4.2 (3rd paragraph):
"*It is noteworthy that in case PISM was used for the glaciers currently modeled with IGM (class 1), the MAE would increase to 42.7 m (IGM: 38.0 m), RMSE to 54.1 m (IGM: 50.1 m) and R would drop to 0.71 (IGM: 0.77). For this comparison, PISM results on the 500-m resolution grid were reprojected to the 100-m resolution IGM grid using nearest neighbor interpolation. The above confirms that the use of IGM for small glaciers leads to better agreement with thickness measurements. One reason may be the higher-order physics behind IGM, which helps to resolve small-scale ice flow and bed features better than with a model like PISM which is based on shallowness assumptions (i.e. small depth-to-width ratios are less likely to apply to glaciers in class 1).*"

In Sect. 4.3 (first paragraph) the following was added/reformulated:
"*By applying dedicated inverse methods and model physics for different glacier types, using state-of-the-art remote sensing and model input datasets, and calibrating against thickness observations, we limit uncertainties in the final thickness and bed maps. Arguably, using different ice flow models, spatial resolution, and individual parameter calibration per glacier class, causes some consistency between the methods to be lost. However, advantageously we achieve a lower misfit with thickness observations by treating glacier types separately. More specifically, the superior performance of IGM for glaciers in class 1, as well as the improved results with PISM for glaciers in class 2, were the main reasons to use two different ice flow models for these classes.*"

and

"*In summary, our modelling choices led to more detailed bed and thickness maps that are in closer agreement with observations, yet at the expense of some coherency.*"

PROGRESS

In response to this comment, you raise the argument that your approach 'can be used as a numerically stable spin up state for prognostic modeling'. I agree that this initialisation is big asset. Yet the built-in IGM inversion for thickness and ice-flow parameters claims the same property. Please discuss. Do not misunderstand me, I can accept this argument as a clear benefit. Yet, I imagine myself to start modelling on Svalbard with your thickness map. I would then need to use IGM and PISM to do so consistently. It might be me, but it seems impractical to do so with two models that need to communicate/interact. Again, the solution is to reduce the method mix to PISM-only (with perfect plasticity).

Thanks for this comment. Admittedly, it would be more work to perform spin-up and subsequent prognostic modeling with two models rather than with one model (more or less twice the amount of work). In the way we have currently chosen to categorize the glaciers, with glaciers in class 1 (small land-terminating glaciers) not having shared boundaries with glaciers in class 2 (large glaciers and ice cap systems) we however disagree that communication or interaction is necessary between the models. One problem we do envision though is for glaciers in class 3 (surging glaciers). Since neither PISM or IGM was used for those glaciers, no spin-up has been done for those glaciers and starting a forward simulation in the present day with an arbitrary ice flow model for those glaciers would likely lead to a 'shock', i.e. sudden geometric changes, at the start of a prognostic run. We have currently no suitable solution for this other than trying to use either PISM or IGM instead for these glaciers as well when preparing for a future simulation.

We added the following to Sect. 4.2 (final paragraph):
"... has the advantage that it can be used as a numerically stable spin up state for prognostic modeling. This currently however only applies to glaciers in classes 1 and 2, for which iterative inverse methods were used. In case also glaciers in class 3 are to be included in a prognostic run, we would suggest to instead use PISM also for these glaciers to allow for spin up and transient forward modelling (as for glaciers in class 2). This inevitably does introduce larger uncertainty in the basal topography and initial ice thickness."

And yes, it is correct that the built-in IGM inversion also can be used to generate a stable spin-up state for prognostic modelling. Similarly, several other inversion methods can be set up to do so as well (e.g. Farinotti et al. 2009) if one aligns the prognostic modelling methodology with the inversion workflow (e.g. same assumptions on ice flow physics, same grid). However, among the so-far existing thickness products for Svalbard, none have been generated in that fashion until now. For example, the Fürst et al. (2018) thicknesses in fast flowing areas were derived using mass-conservation and observed velocities (without any ice flow model), meaning that these thicknesses cannot be used for prognostic simulations without spin-up. This is why we name this benefit in sect. 4.2. With that said, we do not think that it would add much to include more on how other inversion methods could be set up to achieve the same goal.

PERFECT PLASTICITY

Your answers to my minor comments on L91 and L107-108 are not yet satisfying. You agreed that temporal consistency motivated your choice - specifically with regard to the 2010

DEM and 2017-2018 surface velocity observations. Following this logic, you should rather use the Copernicus DEM. Moreover, the perfect plasticity does not require surface velocities. So your argument should invoke the glacier outlines and the DEM. This also corroborates your argument on limiting the surge-type identification to the velocity fields from 2017-18. As I indicated before, the analysis of Koch et al. (2023) shows an extended time coverage for surge-type glacier identification. Finally, you did not comment on the applicability of the perfect plasticity approach for surge-type glaciers. Please add.

We are grateful for these comments. Temporal consistency was one of the factors considered for input data selection. Another one was the quality of the input data. For the DEM, we chose the NPI S0 Terrengmodel which is a dedicated product for Svalbard based on aerial photos and with high spatial resolution. More details are available here: https://data.npolar.no/dataset/dce53a47-c726-4845-85c3-a65b46fe2fea. We used the 20-m resolution Svalbard-wide model here, which is derived from sub-region models at 2-5 m spatial resolution. This DEM is widely used in many studies on Svalbard, e.g. to quantify surface height change (1936-2010) in a recent study in Nature by Geyman et al. (2022). Other DEMs, such as the Copernicus DEM and the ArcticDEM have not been specifically optimized for Svalbard. This was the main reason for us to choose the NPI DEM. Besides that, the highest resolution Copernicus DEM for Europe (EEA-10) does unfortunately not include Svalbard in its domain, hence we would have to use a global version (GLO-30 or GLO-90) instead which has coarser spatial resolution (30 or 90 m). Furthermore, the Copernicus DEM is from 2011-2015, which would still be a 4-5 year mismatch with the timing of the velocity dataset. The following was added to Sect. 2:
"*For surface heights, we chose to use the S0 Terrengmodel by the Norwegian Polar Institute (NPI, 2014), which is a 20-m resolution digital elevation model (DEM), based on aerial photos between 2009-2012 and derived from subset models (5-m resolution) for regions in Svalbard.*"

Our main aim was to select glaciers that due to surging would be affected strongly by the mismatch in timing of the input datasets (velocity, dh/dt and DEM). We use the timing of the velocity dataset to select which glaciers are problematic to model using iterative inverse methods. The fact that the alternative method (perfect plasticity) does not require velocity information seems irrelevant to us. In case class 3 would not exist as a separate class, most surging glaciers would fall in class 2 and would hence be modeled with PISM instead, in which case the inversion would rely heavily on the velocity data to invert for friction. This would create major errors in the thickness inversion when velocities are strongly overestimated (as is the case when a surge happened in 2017-2018 while the DEM is from ~2011 and the dh/dt dataset from ~2015). Arguably we could alternatively have selected surging glaciers based on the timing of the DEM (2009-2012) or dh/dt (2010-2019) datasets instead. However, since the Koch et al. (2023) dataset does not go back further than September 2015, we would have trouble selecting all glaciers that surged during the periods of the dh/dt and DEM datasets. Finally, we would like to note that all 13 glaciers that Koch et al. (2023) listed as actively surging between late 2015 and 2018 were treated as surging glaciers in our approach and were hence modeled with the perfect plasticity assumption. We in fact mistakenly classified Nathorstbreen as surging, which is the only glacier that surged in 2015-2016 and not in 2017-2018. To be correct, we now state that all glaciers that according to Koch et al. (2023) surged between 2015 and 2018 are treated as surging glaciers and included in class 3.

The suitability of the perfect-plasticity method for surging glaciers is hard to verify. We do know from tests that the iterative inverse methods (using PISM and IGM) would introduce major outliers in the thickness map. This was the initial reason to search for an alternative method for these glaciers. A benefit of using the perfect plasticity method is that (in our case) it uses a DEM that is from before the surges initiated, which makes that the thickness inversion corresponds to the quiescent phase. We think this is an advantage because the strongly transient physics and stresses involved in an active surge may not be well captured by the perfect plasticity assumption (or any other ice flow model). We add the following in Sect. 3.3:

"*In the perfect plasticity assumption ice thickness is controlled primarily by the surface height (Eq. 1). Since the DEM (2009-2012) was collected prior to the initiation of the surge for the selected glaciers, the thickness estimation is effectively based on the pre-surge glacier geometry. We regard this as an advantage as ice flow models in general are not well able to describe the strongly transient stress-state of actively surging glaciers.*"

MINOR COMMENTS
L126 'This comment was undressed (neither in the rebuttal nor in the revised manuscript).' From my understanding, the term yield stress relates to when deformation becomes possible whereas the term sliding law normally relates basal velocities to general basal stress conditions. Please check this terminology and be specific about any assumptions.

Good point. Indeed the term yield stress refers to the basal stress required for sliding to occur. This applies to the plastic Coulomb sliding model. Since we rather use a linear sliding law in PISM, the term yield stress may be a less appropriate term to use. We hence change this and now use the term sliding coefficient (C) instead when referring to the sliding law used in PISM. We still use the term "yield constant" in the description of the perfect plasticity assumption, where we think the terminology is appropriate.

L270 'Follow-up on your answer to:' For the land- and marine terminating glaciers, you report mean thickness values of 42 m and 162 m, respectively. How can you reconcile this value with the archipelago-wide average of 205 m. I certainly miss something here ;o) Obviously, I missed the difference between median and mean values. This metric-mix makes it very hard to compare. Solutions would be to either use both metrics for each time you specifically provide averaging measures or only give either of the two metrics consistently throughout the manuscript. Personally, I prefer the former. Your decision.

The large thickness difference is not because of the use of median or mean, it is rather because different quantities are compared. The mean thickness of 205 m is calculated by dividing the total glacier volume by the total glacier area. The median thicknesses for land-terminating glaciers (42 m) and tidewater glaciers (162 m) are the median values of average glacier thicknesses (per glacier) for all 1363 land-terminating glaciers and 181 tidewater glaciers, respectively. In other words, the median values are found by sorting the glacier-average thicknesses for the glaciers in one class from low to high, and then taking the midpoint value (682nd value for land-terminating glaciers; 91st value for tidewater glaciers). These median values are hence strongly affected by the size distribution of glaciers in the RGI dataset. The relatively high number of small (thin) glaciers in both classes

explains the discrepancy between the Svalbard wide mean thickness and the median thickness per glacier class. Anyway, we understand the confusion and have added the following to Sect. 4.1:

"*These median values are much lower than the Svalbard-wide mean ice thickness (205 m), which results in a skewed size-distribution with predominantly small and thin glaciers in both glacier categories (LT and TW).*"

and to the caption of Fig. 5:

"*The box-plots for LT and TW glaciers are based on mean thickness values for every glacier.*"